# Multi-Dimensional Conformal Prediction

**Yam Tawachi & Bracha Laufer-Goldshtein**
School of Electrical and Computer Engineering
Tel-Aviv University
Tel-Aviv, Israel
{yamtawachi@mail,blaufer@tauex}.tau.ac.il

## Abstract

Conformal prediction has attracted significant attention as a distribution-free method for uncertainty quantification in black-box models, providing prediction sets with guaranteed coverage. However, its practical utility is often limited when these prediction sets become excessively large, reducing its overall effectiveness. In this paper, we introduce a novel approach to conformal prediction for classification problems, which leverages a multi-dimensional nonconformity score. By extending standard conformal prediction to higher dimensions, we achieve better separation between correct and incorrect labels. Utilizing this we can focus on regions with low concentrations of incorrect labels, leading to smaller, more informative prediction sets. To efficiently generate the multi-dimensional score, we employ a self-ensembling technique that trains multiple diverse classification heads on top of a backbone model. We demonstrate the advantage of our approach compared to baselines across different benchmarks. [1]

## 1 Introduction

Deep learning models become increasingly dominant in almost every domain, ranging from computer vision and natural language processing to speech recognition. However, as deep learning models are deployed in safety-critical applications, such as healthcare (Lambert et al., 2024) and autonomous driving (Muhammad et al., 2020), it is important to certify their reliability and safety. This highlights the need for robust uncertainty quantification methods that can determine when models are uncertain about their predictions and suggest alternative estimates. Conformal prediction offers a powerful, distribution-free, and model-agnostic framework for uncertainty quantification, providing finite-sample guarantees (Vovk et al., 2015; Lei et al., 2013; Barber et al., 2021; Angelopoulos et al., 2020). Its core principle is to transform pointwise predictions from any model into prediction sets or intervals that contain the true value with high probability.

Conformal prediction relies on a nonconformity score that quantifies how unusual or atypical a new input-label pair is relative to the given data. While conformal prediction guarantees valid coverage for any model and data distribution, its practical effectiveness is influenced by both the performance of the model and the choice of the nonconformity score (Romano et al., 2020; Angelopoulos et al., 2020). The efficiency of the resulting prediction sets is typically measured by their size, with smaller sets leading to more informative and precise predictions. Consequently, there is active research aimed at developing methods that produce the most efficient and concise prediction sets. This includes proposing new nonconformity scores (Sadinle et al., 2019; Romano et al., 2020; Angelopoulos et al., 2020; Huang et al., 2024; Luo & Zhou, 2024a), training models using loss functions that promote efficiency (Stutz et al., 2021; Einbinder et al., 2022), and combining different models, scores, or data augmentations (Bai et al., 2021; Luo & Zhou, 2024b; Lu, 2023).

The common approach across existing methods is to optimize a single nonconformity score, while the calibration process—generally involving the computation of a threshold based on a quantile of the calibration scores—remains unchanged. In this paper, we present a novel perspective that extends the standard conformal prediction framework from a one-dimensional score to a higher-dimensional space defined by multiple nonconformity scores. The intuition behind this

---

[1]Our code is available at: https://github.com/yamtawa/Multi-CP

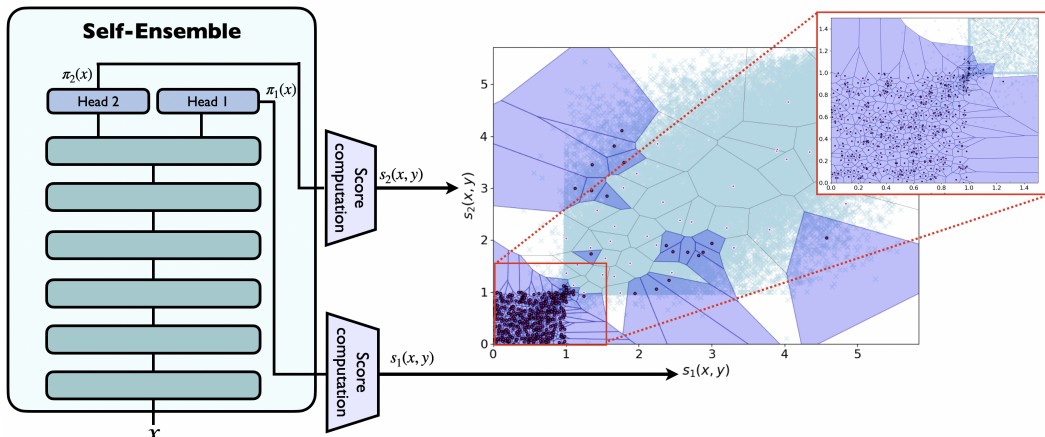

Figure 1: A demonstration of the proposed multi-score conformal prediction for the 2-dimensional case. On the left, an illustration of our proposed self-ensemble model with two classification heads. We compute a nonconformity score for each head. On the right, the 2-dimensional score space is presented, where red circles correspond to scores of true labels, and light-blue x-marks correspond to scores of false labels. Selected cells are colored in blue and their centers have black edge color. We see that cells with low number of false labels are chosen.

is that higher-dimensional spaces can better separate correct from incorrect labels, potentially leading to more efficient prediction sets with fewer false labels. However, selecting a region in this higher-dimensional space that guarantees exact coverage while optimizing efficiency is non-trivial, as there are infinitely many ways to partition the space. To tackle this challenge, we propose a simple yet effective method that splits the multi-dimensional score space into cells, with cell centers defined by the calibration samples, as illustrated in Fig. 1. The cells are ranked bottom-up based on the ratio of incorrect to correct labels that they contain. Then, we select the top-ranked cells that meet the desired coverage. At test time, the prediction set consists of all labels with scores falling within the selected region. Additionally, we introduce a flexible self-ensembling technique that constructs a multi-dimensional score by combining predictions from multiple diverse classification heads built on top of a single backbone model. We provide theoretical guarantees that this high-dimensional region selection maintains valid coverage and show that it is equivalent to optimizing the size of the prediction set, under coverage constraint. Extensive experimental results, demonstrate that our multi-dimensional framework offers superior efficiency compared to baseline methods across various settings.

Our main contributions can be summarized as follows:

1. Propose a new multi-dimensional conformal prediction framework that can better identify regions with a large amount of true labels and a small amount of false labels. Our approach is parameter-free, requires no optimization procedures, and is not restricted to a specific coverage level.
2. Theoretically show that this high-dimensional selection procedure maintains the desired coverage with finite-sample guarantees.
3. Present a flexible and cheap self-ensemble approach to obtain multi-dimensional nonconformity scores by training multiple classification heads, while encouraging diversity.
4. Our experimental results demonstrate the superiority of the proposed method over competing baselines, consistently producing smaller and more efficient prediction sets.

## 2 RELATED WORK

**Enhanced nonconformity scores.** Improving the efficiency of conformal prediction has been a central focus of recent research. Several studies have derived enhanced nonconformity scores aimed at reducing the size of the prediction sets or improving conditional coverage (Sadinle et al., 2019; Romano et al., 2020; Angelopoulos et al., 2020; Huang et al., 2024; Luo & Zhou, 2024a). Another approach involves performing conformal prediction in the feature space, then mapping the intervals from the embedding space back to the output space (Teng et al., 2022). Our approach addresses the use of multiple nonconformity scores, as previous research has demonstrated that combining multi-

ple scores can be more efficient than relying on a single score (Yang et al., 2023b). We introduce a practical method for generating multiple scores without the need for training multiple models or performing additional inference steps. This approach can be applied to any base score and is orthogonal to the advancements in developing improved nonconformity scores.

**Improving conformal prediction via training.** Although conformal prediction is usually considered as a wrapper around black-box models, recent approaches suggested to directly train models to improve conformal prediction efficiency. Stutz et al. (2021) introduced a differentiable conformal prediction pipeline that optimizes the size of the prediction sets. In (Einbinder et al., 2022), a regularization term was added to the training loss, encouraging the distribution of the nonconformity scores to match a uniform distribution. Additionally, Bai et al. (2021) explored optimizing conformal prediction within broader function classes. These methods rely on differentiable approximations that may not fully align with the actual target objective, and are often designed for a specific coverage level, requiring retraining for each new level. In contrast, our approach avoids optimization altogether and offers greater flexibility by being independent of the coverage level and the base nonconformity score.

**Combining nonconformity scores.** Several works explore conformal prediction in the context of model fusion and combining nonconformity scores. Ensemble learning methods, which train multiple models on different subsets of the data, have been widely applied to conformal prediction (Linusson et al., 2020). Notable approaches include cross-conformal predictors (Vovk, 2015), bootstrap conformal predictors (Vovk, 2015), and out-of-bag calibrated conformal predictors (Devetyarov & Nouretdinov, 2010). Other methods for combining nonconformity scores have also been investigated. For example, Luo & Zhou (2024b) proposed using a weighted average of multiple scores derived from the same output, with weights learned through optimization. Similarly, Lu (2023) suggested combining nonconformity scores obtained via test-time augmentations of the same image. A more recent work explored the aggregation of multiple prediction sets, assuming no direct access to the underlying scores (Gasparin & Ramdas, 2024). In contrast to existing approaches that focus on combining nonconformity scores via weighted aggregation or majority voting, we propose a general framework that identifies promising regions with low concentrations of false labels in the multi-dimensional score space.

**Ensemble methods.** Model ensembles have been shown to enhance various metrics for uncertainty quantification beyond conformal prediction efficiency, such as calibration error (Hansen & Salamon, 1990; Lakshminarayanan et al., 2017). However, ensembles are often considered computationally expensive, as they require training and deploying multiple independent models. To mitigate this, Qendro et al. (2021) proposed an early-exit ensemble, which leverages multiple prediction heads from intermediate layers to improve uncertainty quantification while maintaining a single model. This approach has been shown to enhance computational efficiency (Cai et al., 2020) and improve adversarial robustness (Qendro & Mascolo, 2022). Building on these insights, we propose a self-ensemble model with multiple classification heads to generate a multi-dimensional nonconformity score without significant additional costs.

## 3 BACKGROUND - CONFORMAL PREDICTION

Let $X \in \mathcal{X}$ represent an input, associated with a label $Y \in \mathcal{Y}$, where $\mathcal{Y} = \{1, \ldots, Q\}$. Consider a classifier $\pi(x) \in [0,1]^Q$ that outputs a probability distribution over $Q$ classes (e.g., a neural network with a softmax layer, producing probabilities for each class). Conformal prediction starts by computing a *nonconformity score* $s : \mathcal{X} \times \mathcal{Y} \to \mathcal{S} \subseteq \mathbb{R}$, which quantifies the uncertainty of the classifier's prediction for the pair $(X, Y)$ with respect to existing data. Given a set of calibration points $\{(X_i, Y_i)\}_{i=1}^m$, we can form a prediction set for a new test point $X_{m+1}$. Provided that the points $\{(X_i, Y_i)\}_{i=1}^{m+1}$ are exchangable, this set is constructed to have at least $1 - \alpha$ coverage of the true label, where $\alpha \in (0, 1)$ is set by the user. The prediction set is defined as:

$$\Gamma_\lambda(X_{m+1}) = \{y \in \mathcal{Y} : s(X_{m+1}, y) \leq \lambda\}, \tag{1}$$

where the threshold $\lambda$ is set to the quantile of the calibration nonconformity scores:

$$\lambda := \text{Quantile}\left(\frac{\lceil (m+1)(1-\alpha) \rceil}{m}; \{s(X_i, Y_i)\}_{i=1}^m\right). \tag{2}$$

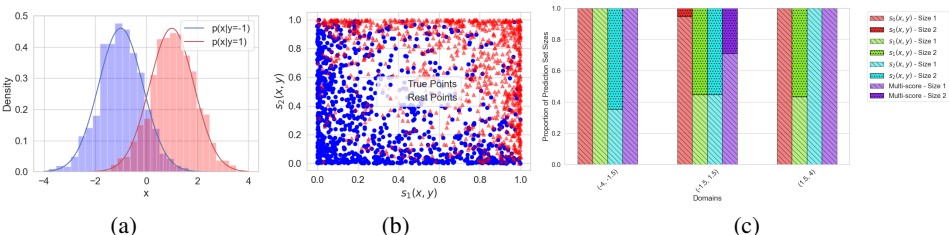

Figure 2: Binary classification example. (a) $\mathbb{P}(x|y)$ for $y = \{-1, 1\}$. (b) The 2-dimensional score space defined by $s_1(x, y)$ and $s_2(x, y)$. (c) The set-sizes obtained for different $x$ domains based on $s_0(x, y)$, $s_1(x, y)$, $s_2(x, y)$ and our multi-score method using both $s_1(x, y)$ and $s_2(x, y)$.

This procedure ensures that the true label $Y_{m+1}$ is included in the prediction set with at least $1 - \alpha$ probability, as stated in the following theorem.

**Theorem 1.** *(Conformal calibration coverage guarantee) .Let $\{(X_i, Y_i)\}_{i=1}^{m+1}$ be exchangeable data points. For any score function $s : \mathcal{X} \times \mathcal{Y} \rightarrow \mathcal{S}$ and any significance level $\alpha \in (0, 1)$, define the quantile $\lambda$ by Eq. (2) and the prediction set $\Gamma_\lambda(X_{m+1})$ by Eq. (1). We have:*

$$\mathbb{P}(Y_{m+1} \in \Gamma_\lambda(X_{m+1})) \geq 1 - \alpha. \tag{3}$$

Commonly used nonconformity scores are described in Appendix A.1.

## 4 PROPOSED METHOD

### 4.1 MULTI-SCORE CALIBRATION

We now consider a multi-dimensional nonconformity score $\mathbf{s} : \mathcal{X} \times \mathcal{Y} \rightarrow \mathcal{S}$, constructed by concatenating $n$ individual nonconformity scores as $\mathbf{s}(x, y) = [s_1(x, y), \dots, s_n(x, y)]^T$, where $\mathcal{S} := \mathcal{S}_1 \times \cdots \times \mathcal{S}_n \subseteq \mathbb{R}^n$. Before discussing different approaches to handling multi-dimensional nonconformity scores and presenting our proposed method, we first introduce a toy example to highlight the advantages of using a multi-dimensional score over a single-dimensional one.

**Example 4.1** (Toy setting)**.** Assume a binary classification problem with $\mathcal{Y} = \{-1, 1\}$ and prior probabilities $\mathbb{P}(Y = 1)$ and $\mathbb{P}(Y = -1)$. The input $X$ is generated from a mixture of two Gaussians, with $\mathbb{P}(X|Y) = \mathcal{N}(Y, \sigma_y^2)$. In this example, we can compute the posterior $\mathbb{P}(Y|X)$ using Bayes rule. Let $\mathbb{P}(y|x) = \mathbb{P}(Y = y|X = x)$, we consider the following three classifiers:

$$\pi_0(x) = \mathbb{P}(y|x), \ \pi_1(x) := \begin{cases} \mathbb{P}(y|x), & \text{if } x < 0, \\ (\epsilon, 1 - \epsilon), & \text{if } x > 0 \end{cases}, \ \pi_2(x) := \begin{cases} (\epsilon, 1 - \epsilon), & \text{if } x < 0, \\ \mathbb{P}(y|x), & \text{if } x > 0 \end{cases} \tag{4}$$

where $\epsilon \sim \mathcal{N}(0, 1)$. Here, $\pi_0(x)$ is the ideal classifier, while $\pi_1(x)$ and $\pi_2(x)$ are identical to the ideal classifier over half the range of $X$, but uninformative over the remaining half. Let $s_i(x, y)$ denote the nonconformity score computed over the $i$-th classifier, it is clear that performing conformal prediction over $s_0(x, y)$ would be the most efficient, however, using either $s_1(x, y)$ or $s_2(x, y)$ will lead to suboptimal results in the uninformative regions. Performing conformal prediction in the 2-dimensional space, defined by $\mathbf{s}(x, y) = [s_1(x, y), s_2(x, y)]^T$, is advantageous as the individual scores provide complementary information for different ranges of the input $X$. This can be seen from Fig. 2 (b), where taking both axes into account can help to identify regions with high density of true points versus false points. Setting $\alpha = 0.1$, we obtain the following average set sizes: 1.05 for $s_0(x, y)$, 1.48 for $s_1(x, y)$, 1.47 for $s_2(x, y)$ and 1.24 for our proposed method. Figure 2 (c) compares the set sizes per $x$-domain. As expected, $s_0(x, y)$ produces two-element sets, only in the middle region, where the Gaussians overlap. In contrast, $s_1(x, y)$ and $s_2(x, y)$ produce a higher proportion of two-element sets in the corresponding noisy regions. Our method, which relies on both scores, closely matches the behavior of the ideal score. Further details are provided in Appendix A.2.

We now turn to the analysis of the multi-dimensional score space. The construction of such scores is discussed in § 4.4. In standard conformal prediction, where $n = 1$, the threshold $\lambda$ in Eq. (2) divides the real line into two regions: scores less than or equal to $\lambda$ are included in the prediction

set, while scores greater than $\lambda$ are excluded. This simple thresholding approach is effective because nonconformity scores are expected to be low for true labels, which conform to the patterns in the data, and high for false labels, which deviate from the expected behavior, as explained in § 3. When $n > 1$, we have a multi-dimensional score. Intuitively, scores that are close to the minimum value in all dimensions correspond to the most conforming labels, while increasing any component of $\mathbf{s}(x, y)$ leads to less conforming scores. However, unlike the one-dimensional case, it is not immediately clear how to optimally partition the score space $\mathcal{S}$ into regions that should be included or excluded from the prediction set.

A common approach for handling multiple nonconformity scores is to use their weighted sum:

$$s_{\mathrm{w}}(x, y) = \sum_{i=1}^{n} w_i s_i(x, y), \tag{5}$$

where the weights $w_i$ are optimized to minimize the size of the prediction set while maintaining the desired coverage level (Bai et al., 2021; Luo & Zhou, 2024b). Geometrically, this is equivalent to splitting the score space $\mathcal{S}$ with a hyperplane of the form $w_1 s_1(x, y) + \cdots + w_n s_n(x, y) = \lambda$, and including only those scores that lie below this hyperplane, i.e. scores satisfying $w_1 s_1(x, y) + \cdots + w_n s_n(x, y) \leq \lambda$. However, this method imposes a rigid structure on the partitioning, which may not align with the optimal regions that yield the smallest possible prediction sets. Moreover, it requires tuning the weights for a specific coverage level, making it less flexible and potentially unsuitable for different values of $\alpha$.

We propose a more flexible approach for managing the multi-dimensional score space that eliminates the need for weight optimization. Our method begins by partitioning the calibration data into two disjoint subsets: $\mathcal{D}_{\mathrm{cal}} = \mathcal{D}_{\mathrm{cells}} \cup \mathcal{D}_{\mathrm{re\text{-}cal}}$, where $\mathcal{D}_{\mathrm{cells}} = \{(X_i, Y_i)\}_{i=1}^{k}$ and $\mathcal{D}_{\mathrm{re\text{-}cal}} = \{(X_i, Y_i)\}_{i=k+1}^{m}$, with $m - k = r$. The subset $\mathcal{D}_{\mathrm{cells}}$ is used for partitioning the score space $\mathcal{S}$ into distinct regions (cells) and evaluating the quality of each region. Next, the subset $\mathcal{D}_{\mathrm{re\text{-}cal}}$ is utilized for calibrating the total selection region, ensuring that the resulting prediction sets achieve the desired coverage level. Thus, our method consists of three main stages: (i) partitioning, (ii) scoring and ranking, and (iii) calibration, as detailed below.

**(i) Partitioning.** We aim to partition the score space $\mathcal{S}$ into cells and later decide which cells to include in the prediction sets. Using a uniform grid is computationally expensive and not scalable with increasing number of dimensions $n$. Additionally, the score distribution is typically uneven, with some regions being densely populated and others sparse, making uniform partitioning inefficient. Instead, we partition $\mathcal{S}$ into $k$ cells, centered at $\mathbf{s}(X_1, Y_1), \ldots, \mathbf{s}(X_k, Y_k)$ for all samples in $\mathcal{D}_{\mathrm{cells}}$. Specifically, each point in $\mathcal{S}$ is assigned to the closest center:

$$\mathcal{C}_i = \left\{ \tilde{\mathbf{s}} \in \mathcal{S} \,\middle|\, i = \arg\min_{1 \leq j \leq k} \|\tilde{\mathbf{s}} - \mathbf{s}(X_j, Y_j)\| \right\}, \quad i \in \{1, \ldots, k\}. \tag{6}$$

This way, the cell resolution adapts to the score density, with smaller cells in high-density regions and larger cells in low-density areas. Note also that this approach is analogous to standard conformal prediction, where the calibration scores define segments of varying lengths, and the final selected interval is the union of all segments to the left of the computed quantile (2). Thus, our cell partitioning method can be seen as a generalization of the standard partitioning process to higher dimensions.

**(ii) Scoring and ranking.** In the next stage, we determine which cells to include in the prediction sets by computing a ratio that quantifies the balance between false and true labels within each cell. Let $F_i = \sum_{j=1}^{k} \sum_{q=1}^{Q} \mathbf{1}\{\mathbf{s}(X_j, q) \in \mathcal{C}_i\} \cdot \mathbf{1}\{Y_j \neq q\}$ and $T_i = \sum_{j=1}^{k} \mathbf{1}\{\mathbf{s}(X_j, Y_j) \in \mathcal{C}_i\}$ denote the number of scores with false labels and true labels, respectively. We define the cell scores $D_i$ as follows:

$$D_i := \frac{F_i + T_i}{T_i} = \frac{F_i}{T_i} + 1, \quad i \in \{1, \ldots, k\}. \tag{7}$$

This ratio reflects the relative amount of false to true labels within each cell $\mathcal{C}_i$. We normalize by the number of true scores in each cell, accounting for the possibility of multiple identical true scores (overlapping cells), though this occurrence becomes increasingly rare as $n$ increases. Note that a definition of $D_i' = \frac{F_i}{T_i}$ results in the same score ordering. However, we keep the definition of Eq. (7), as it is essential for showing the equivalence between our scoring and ranking procedure and solving the set-size optimization problem, described in § 4.2.

In order to improve conformal prediction efficiency and obtain smaller set sizes at test time, we prioritize regions with a low false-to-true label ratio while avoiding regions with high false-to-true ratio. Let $k'$ denote the number of unique cells, where multiple identical scores are treated as a single cell. We then rank the sequence of unique cells $\mathcal{C}_{(1)}, \mathcal{C}_{(2)}, \ldots, \mathcal{C}_{(k')}$ according to $D_i$, from lowest to highest, i.e., $D_{(1)} \leq D_{(2)} \leq \cdots \leq D_{(k')}$.

**(ii) Calibration.** The final step is selecting the regions that will ensure exact coverage. In this stage, we utilize the re-calibration set $\mathcal{D}_{\text{re-cal}}$. We start from the cell with lowest $D_i$ score, and progressively add cells until the desired coverage is achieved over $\mathcal{D}_{\text{re-cal}}$. Formally, we define the selected region $\mathcal{C}_{\text{in}}^{\eta}$ as the union of the top-ranked cells up to index $\eta$:

$$\mathcal{C}_{\text{in}}^{\eta} = \bigcup_{i=1}^{\eta} \mathcal{C}_{(i)}.$$

The required $\eta$ is determined by:

$$\eta^* = \min \left\{ \eta \in \{1, \ldots, k'\} \mid Y_i \in \mathcal{C}_{\text{in}}^{\eta} \text{ for at least } \lceil (1-\alpha)(r+1) \rceil \text{ samples } (X_i, Y_i) \in \mathcal{D}_{\text{re-cal}} \right\}, \tag{8}$$

and $\mathcal{C}_{\text{in}}^{\eta^*}$ is the final selected region. It is essential to use a separate calibration set rather than reusing $\mathcal{D}_{\text{cells}}$. This is because the cell scores are derived from $\mathcal{D}_{\text{cells}}$, and selecting cells based on the same data would introduce bias, ultimately leading to undercoverage (see Appendix A for further justification).

At test time, given a new test point $X_{m+1}$ with an unknown label $Y_{m+1}$, we compute the multi-dimensional score $\mathbf{s}(X_{m+1}, y)$ for each possible $y \in \mathcal{Y}$. We then include only the labels that fall within the selected region $\mathcal{C}_{\text{in}}^{\eta^*}$, yielding the following prediction set:

$$\Gamma_{\eta^*}(X_{m+1}) = \left\{ y \in \{1, \ldots, Q\} \mid \mathbf{s}(X_{m+1}, y) \in \mathcal{C}_{\text{in}}^{\eta^*} \right\}. \tag{9}$$

Our method, Multi-Score Conformal Prediction, is summarized in Algorithm 1. Unlike the single-score thresholding approach (1) or the hyperplane splitting for weighted scores (5), our approach defines an unstructured selection region in the multi-dimensional score space. This flexibility enables us to prioritize regions with fewer false labels, leading to smaller prediction sets. Notably, our method differs from vector quantile regression (VQR), which extends quantile regression to multivariate settings (Carlier et al., 2016; Feldman et al., 2023; Rosenberg et al., 2022). While VQR captures the central $1 - \alpha$ portion of the output distribution, our approach focuses on optimizing the selection region to minimize set size while ensuring valid coverage.

### 4.2 THEORETICAL ANALYSIS

Although our construction seems different from standard conformal prediction, it still provides coverage guarantees, as stated in the following proposition.

**Proposition 2.** *(Multi-score conformal calibration coverage guarantee). Let $\mathcal{D}_{\text{cells}} = \{X_i, Y_i\}_{i=1}^{k}$ and $\mathcal{D}_{\text{re-cal}} = \{X_i, Y_i\}_{i=k+1}^{m}$ be two disjoint datasets, and the samples $\{(X_i, Y_i)\}_{i=k+1}^{m+1}$ are exchangeable. For any multi-dimensional score function $\mathbf{s} : \mathcal{X} \times \mathcal{Y} \to \mathcal{S} \subseteq \mathbb{R}^n$ and any significance level $\alpha \in (0, 1)$, the prediction set $\Gamma_{\eta^*}(X_{m+1})$ defined by Eq. (9) satisfies:*

$$\mathbb{P}\left(Y_{m+1} \in \Gamma_{\eta^*}(X_{m+1})\right) \geq 1 - \alpha \tag{10}$$

The proof, detailed in Appendix A.3, is based on defining a mapping function from the multi-dimensional score to the corresponding cell ratio score, defined in Eq. (7). By formulating the predicted set in Eq. (9) using a thresholding operation over this score, our method is aligned with standard one-dimensional conformal prediction, ensuring valid coverage.

We show now that our approach is similar to previous works, optimizing the set size, subject to a coverage constraint (Stutz et al., 2021; Bai et al., 2021; Kiyani et al., 2024). Since the space of all possible prediction sets is overly complex, the problem must be relaxed. Bai et al. (2021) proposed to optimize an arbitrary class of prediction sets $\Gamma_\theta$ parametrized by $\theta$, while Stutz et al. (2021) optimize a parametrized score $s_\theta(x, y)$. Kiyani et al. (2024) considered structured prediction sets $\Gamma^h(x) = \{y \in Y \mid s(x, y) \leq h(x)\}$ with a learned adaptive threshold $h : \mathcal{X} \to \mathbb{R}$. In contrast, we

---

**Algorithm 1** Multi-Score Conformal Prediction

---

**Definitions:** $\mathbf{s}(x, y)$ is a multi-dimensional score function. $\mathcal{D}_{\text{cal}}$ is the calibration data of size $m$. $X_{m+1}$ is a new test sample, $\alpha$ is the miscoverage level, $k$ is the number of samples for cell-partitioning and scoring, and $r = m - k$ is the number of samples for re-calibration.

1: **function** MULTI-SCORE-CP($\boldsymbol{s}(x, y), \mathcal{D}_{\text{cal}}, \alpha$)
2:       Randomly split $\mathcal{D}_{\text{cal}}$ to $\mathcal{D}_{\text{cells}} = \{(X_i, Y_i)\}_{i=1}^{k}$ and $\mathcal{D}_{\text{re-cal}} = \{(X_i, Y_i)\}_{i=k+1}^{m}$
3:       Compute the scores $\mathbf{s}(X_i, Y_i)$, $i \in \{1, \ldots, m\}$
4:       Segment $\mathcal{S}$ into cells $\{\mathcal{C}_i\}_{i=1}^{k}$ centered at $\{\mathbf{s}(X_i, Y_i)\}_{i=1}^{k}$
5:       Compute $D_i, i = 1, \ldots, k$ according to Eq. (7)
6:       Remove duplicate cells $\mathcal{C}_1, \mathcal{C}_1, \ldots, \mathcal{C}_{k'}$
7:       Rank the cells $\mathcal{C}_{(1)}, \mathcal{C}_{(2)}, \ldots, \mathcal{C}_{(k')}$ according to $D_{(1)} \leq D_{(2)} \leq \ldots \leq D_{(k')}$
8:       $\eta^* \leftarrow \min\{\eta \in \{1, \ldots, k'\} \mid Y_i \in \mathcal{C}_{\text{in}}^{\eta} \text{ for at least } \lceil (1-\alpha)(r+1) \rceil \text{ samples in } \mathcal{D}_{\text{re-cal}}\}$
9:       $\mathcal{C}_{\text{in}}^{\eta^*} \leftarrow \cup_{i=1}^{\eta^*} \mathcal{C}_{(i)}$
10:      **return** $\mathcal{C}_{\text{in}}^{\eta^*}$
11: **function** MULTI-SCORE-EVALUATION($\boldsymbol{s}(x, y), X_{m+1}, \mathcal{C}_{\text{in}}^{\eta^*}$)
12:      Compute the scores $\mathbf{s}(X_{m+1}, y)$, $y \in \{1, \ldots, Q\}$
13:      Construct the prediction set $\Gamma_{\eta^*}(X_{m+1}) = \left\{ y \in \{1, \ldots, Q\} \mid \mathbf{s}(X_{m+1}, y) \in \mathcal{C}_{\text{in}}^{\eta^*} \right\}$
14:      **return** $\Gamma_{\eta^*}(X_{m+1})$

---

work in the multi-score domain and consider sets defined as a union of cells in $\mathcal{D}_{\text{cells}}$, i.e. $\Gamma^I(x) = \{y \in \mathcal{Y} | \mathbf{s}(x, y) \in \cup_{i \in I} \mathcal{C}_i\}$ with $I \subseteq 2^k$. We obtain the following relaxed optimization problem:

$$\underset{I \subseteq 2^k}{\arg \min} \, L_{\text{size}} := \mathbb{E}\left[\text{size}(\Gamma^I(X))\right] \quad \text{s.t.} \quad L_{\text{coverage}} := \mathbb{P}\left(Y \in \Gamma^I(X)\right) \geq 1 - \alpha. \quad (11)$$

where $\text{size}(\Gamma^I(X)) = \sum_{q=1}^{Q} \mathbf{1}\{q \in \Gamma^I(X)\}$. In the following lemma, we relate this optimization problem to our defined cell scores (see the proof in Appendix A.4).

**Lemma 3.** *The optimization problem in Eq. (11) admits the following finite sample representation:*

$$\underset{I \subseteq 2^k}{\arg \min} \, \frac{1}{k} \sum_{i \in I} D_i \quad \text{s.t.} \quad \hat{L}_{coverage} := \frac{|I|}{k} \geq 1 - \alpha. \quad (12)$$

It immediately follows that solving Eq. (12) does not require enumerating all possible sets $I$. Instead, we can rank cells according to $D_i$ and select a proportion of $1 - \alpha$ cells with highest ranks. However, this does not guarantee exact coverage, thus, we perform re-calibration by choosing the top-ranked cells covering $\lceil (1-\alpha)(r+1) \rceil$ samples of $\mathcal{D}_{\text{re-cal}}$. This is summarized in the following proposition.

**Proposition 4.** *Algorithm 1 is equivalent to solving a finite sample approximation of Eq. (11), as defined by Eq. (12), followed by a re-calibration stage, ensuring valid coverage (Eq. (10)).*

## 4.3 ADDITIONAL VARIANTS OF MULTI-SCORE CONFORMAL PREDICTION

In the following, we present two additional variants of our proposed method.

**Jackknife+ Multi-Score Conformal Prediction.** A limitation of our approach is that it uses only part of the data to perform the calibration, which may impact efficiency. This is especially critical if the sample size $m$ is small. An alternative solution is to adopt a jackknife+ approach (Romano et al., 2020; Barber et al., 2021), which is computationally more intensive but often provides tighter prediction sets. This approach is summarized in Algorithm B.1. The key idea is to leverage the entire calibration dataset while systematically excluding the $i$th point from both the set of centers and the score computation in Eq. (7). For each possible label $y \in \mathcal{Y}$, we assign it to the nearest center after removing the $i$th center, and similarly, we compute the assignment for the score $s(X_i, Y_i)$. We then compare the ranks of the selected cells and include $y$ in the final prediction if its rank is smaller than $\lceil (1-\alpha)(m+1) \rceil$ hold-out calibration ranks. This ensures a theoretical coverage of $1 - 2\alpha$. However, in practice, the achieved coverage is close to $1 - \alpha$, even without the adjustment $\alpha' = \alpha/2$.

**Soft Multi-Score Conformal Prediction.** We introduce a generalized variant of our approach, replacing hard assignment to the nearest center with a soft assignment to the $b$ nearest neighbors. This variant, outlined in Algorithm B.2, selects the $b$ closest neighbors and includes a point in the prediction set if at least half of them belong to the chosen region.

## 4.4 MULTI-SCORE CONSTRUCTION

Multi-dimensional nonconformity scores can be constructed in several ways. Multiple scores can be obtained from a model ensemble, which can be resource-intensive. Alternatively, test-time augmentations can be used to generate multiple scores for different input augmentations (Lu, 2023). Similarly to ensembling, test-time augmentation requires multiple forward passes, and is generally more appropriate for image data. A more efficient approach is to derive multiple types of scores from a single model output (Luo & Zhou, 2024b). While this requires only a single forward pass, all scores are generated from the same output, which may limit their ability to fully capture predictive uncertainty.

Our multi-dimensional approach is applicable to any of the previously mentioned multi-scores. However, we introduce a novel method for generating diverse nonconformity scores that capture varied perspectives without increasing computational complexity. To achieve this, we attach multiple classification heads, $\{\pi_i(x)\}_{i=1}^n$, to the penultimate layer (second-to-last layer) of the model. Training these heads solely with cross-entropy (CE) loss often leads to highly similar outputs, limiting their usefulness for uncertainty estimation. To mitigate this, we adopt the regularization technique from (Qendro et al., 2021), encouraging diversity by minimizing similarity among the heads. This aligns with the intuition from Example 4.1, where complementary predictions—specialized for distinct input regions—can be combined to enhance uncertainty quantification across the entire range. The classification heads are trained using the following loss function:

$$\mathcal{L} = \frac{1}{n} \sum_{i=1}^n L_{\text{CE}}(\pi_i(x), y) - \frac{\beta}{n(n-1)} \sum_{i=1}^n \sum_{i \neq j} \text{sim}(\pi_i(x), \pi_j(x)), \tag{13}$$

where $L_{\text{CE}}(\cdot, \cdot)$ denotes CE loss, $\text{sim}(\cdot, \cdot)$ denotes cosine similarity, and $\beta$ is a regularization weight, set to 1 in our experiments. While (Qendro et al., 2021) proposed generating classification heads from various layers and depths of the network, our empirical findings revealed that, as expected, heads from shallower layers tend to be weaker, and thus have minimal contribution in the multi-score setting. By attaching the classification heads to the penultimate layer and applying diversity regularization as in Eq. (13), we achieve robust and diverse classification heads.

## 5 EXPERIMENTS

### 5.1 DATASETS AND MODELS

We test our method over three image classification datasets, with varying number of classes and difficulty levels: CIFAR100 (Krizhevsky et al., 2009), Tiny ImageNet (Le & Yang, 2015), and PathMNIST (Yang et al., 2023a). For all datasets we use a ResNet50 backbone model pretrained on ImageNet. We first attach a single classification head and fine-tune the full model with CE loss. Next, we add additional 6 classification heads ($n = 7$ heads in total), freeze the backbone model, and train the heads using the loss defined in Eq. (13). Based on the heads' output probabilities, we compute the nonconformity scores, where we use either regularized adaptive prediction sets (RAPS) (14) or sorted adaptive prediction sets (SAPS) (15) as base scores. We refer to Appendix C for further details on datasets, model architectures, and the training procedure.

### 5.2 EVALUATION

We compare the proposed multi-score method to the following baselines:

- **Best single head (Best head)** - Conformal prediction is applied independently to each classification head using the entire $\mathcal{D}_{\text{cal}}$ dataset, as opposed to our method that performs calibration only over $\mathcal{D}_{\text{re-cal}}$. We report the results for the head that achieves the smallest set size among the evaluated heads.

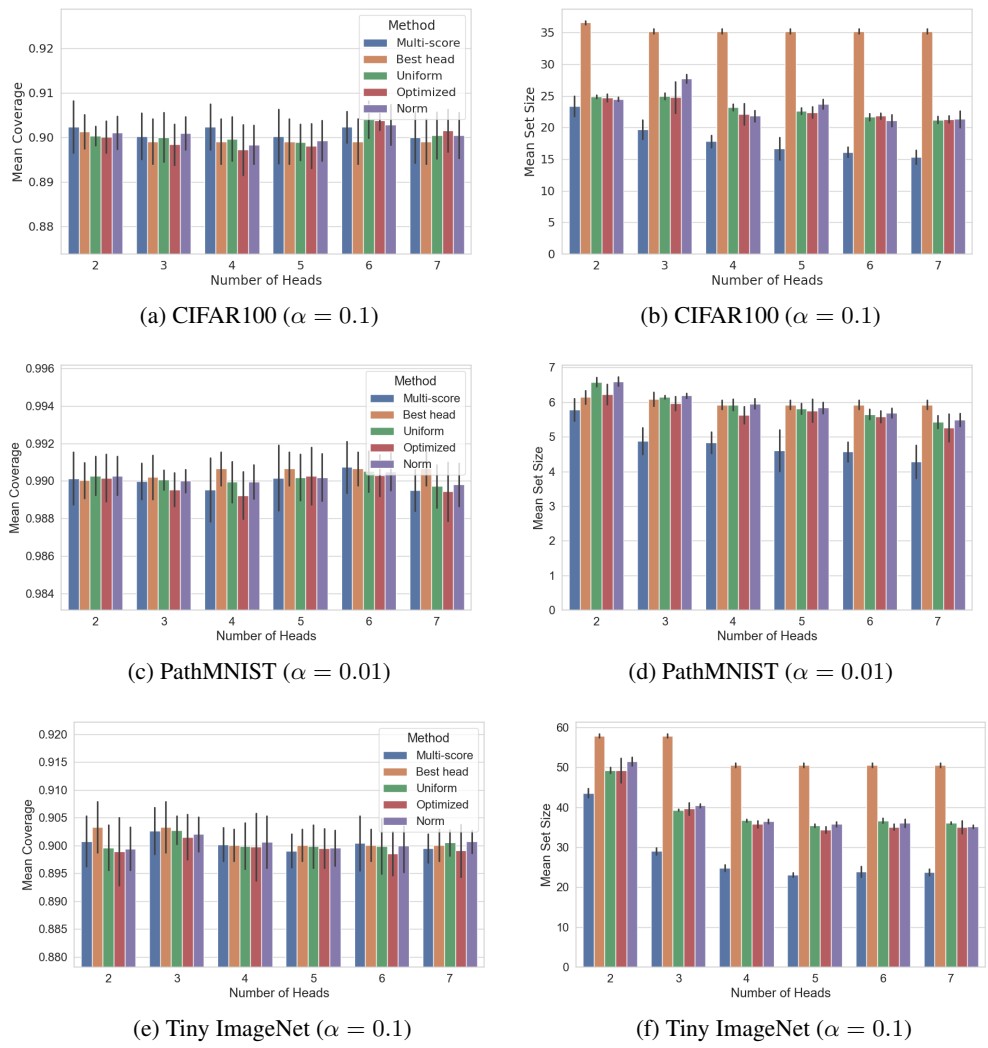

Figure 3: Results for RAPS. Empirical coverage (left), and mean set size (right) as a function of the number of classification heads.

- **Uniform average (Uniform)** - We average the scores obtained for each head and perform standard conformal prediction on the entire $\mathcal{D}_{\text{cal}}$ dataset.
- **Optimized weights (Optimized)** - We use the weighted score $s_{\text{W}}$ defined in Eq. (5). We perform constrained optimization, minimizing the mean set size with a constraint that the empirical mis-coverage does not exceed $\alpha$. The weights are optimized using Optuna (Akiba et al., 2019) over $\mathcal{D}_{\text{cells}}$ with 100 optimization steps. Then, we perform standard conformal prediction over $\mathcal{D}_{\text{re-cal}}$.
- **Norm-based score (Norm)** - Conformal prediction is performed over $\mathcal{D}_{\text{cal}}$ with a norm-based score, defined as $s_{\text{N}}(x, y) = \|\mathbf{s}(x, y)\|_2 = \sqrt{\sum_{i=1}^{n} s_i^2(x, y)}$.

We evaluate the different methods in terms of the empirical coverage $\frac{1}{|\mathcal{D}_{\text{test}}|} \sum_{(X,Y) \in \mathcal{D}_{\text{test}}} \mathbf{1}\{Y \in \Gamma(X)\}$ and the mean set size $\frac{1}{|\mathcal{D}_{\text{test}}|} \sum_{(X,Y) \in \mathcal{D}_{\text{test}}} |\Gamma(X)|$, computed over the test data $\mathcal{D}_{\text{test}}$. We report the average results and the standard deviation over 10 random splits to calibration and test.

## 5.3 RESULTS

**Varying number of heads.** Results as a function of the number of heads are shown in Figs. 3 and D.2, with RAPS and SAPS as base scores, respectively. Results for additional $\alpha$ levels are provided in Figs. D.1 and D.3. As expected, all methods obtain the required coverage. We observe that the proposed method leads to smaller prediction sets, with decreased sizes as the number of

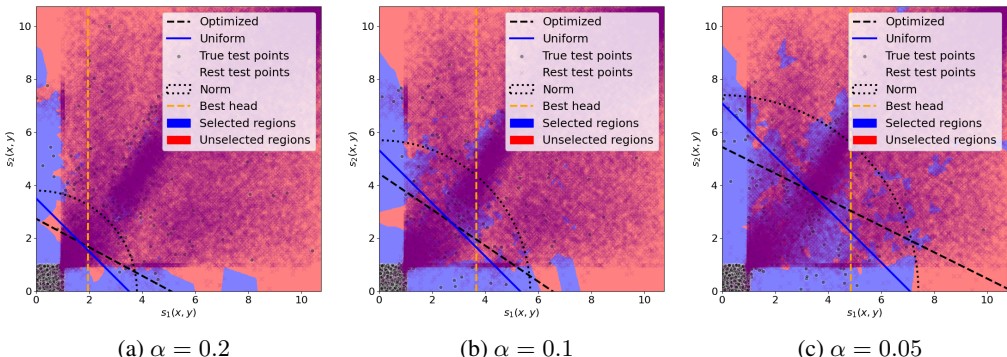

(a) $\alpha = 0.2$       (b) $\alpha = 0.1$       (c) $\alpha = 0.05$

Figure 4: Selection regions for a 2-dimensional score ($n = 2$) using RAPS on Tiny ImageNet at different $\alpha$ levels. Our method's selected region is shaded in blue, while the unselected region is in orange. Baseline decision boundaries are shown as dashed lines, with selected regions lying to the left of the 'Best head' boundary and below the boundaries for all other baselines. True test points are shown as green circles, while those associated with incorrect labels are marked with purple x-marks.

heads increases. Similar trends are observed for both RAPS and SAPS scores. Figure D.7 illustrates how the set size distribution evolves as the number of heads increases, demonstrating the advantage of our method in generating smaller set sizes compared to baselines, especially as $n$ grows.

**Selection region.** Figure 4 demonstrates the results obtained for the 2-dimensional case ($n = 2$). We present the region selected by the proposed method (blue area), and the decision boundaries for the baseline methods. In addition, the test scores for true and false labels are presented. We observe that the test scores are concentrated in two distinct square regions: the bottom-left corner, where true labels dominate, and the top-right corner, where false labels are more prevalent. Our method effectively focuses on regions with fewer false labels, whereas the baselines, constrained by their fixed structure, inevitably include areas with a high density of false labels. Figure D.4 illustrates the cell selection order defined by Eq. (7). Here too, we observe that cells near the bottom left corner are preferable, as well as cells that have low score in either dimension.

**Additional results.** We briefly highlight additional results presented in Appendix D. Tables D.1 and D.2 report the performance in terms of the conditional coverage, with groups defined by either set-size or random projections, respectively. We observe that all methods exhibit similar behavior regarding the maximum coverage violation. Figure D.6 presents the set sizes obtained for Thr and APS scores, highlighting the superiority of our method when temperature scaling is applied to spread out the score distribution. Additionally, we conducted several experiments to demonstrate the versatility of our approach in other multi-score settings, including: (i) a standard ensemble, (ii) test-time augmentation, and (iii) different score types computed for a single head. Moreover, we conducted experiments on text classification (Tab. D.8), and ImageNet (Tab. D.9) verifying that our method is suitable for other data types and large datasets. We evaluated the two additional variants of our method, presented in § 4.3, and found that the Jackknife+ version yields smaller sets, while the soft version with $b > 1$ generally offers no advantage over the basic version with $b = 1$. We also performed an ablation study over the regularization weight $\lambda$ in Fig. D.11, highlighting the benefit of adding diversity regularization. Furthermore, we show that our method is not highly sensitive with respect to the sample size of $\mathcal{D}_{cal}$ (Fig. D.12) and $\mathcal{D}_{cells}$ (Tab. D.4), and maintains robust performance regardless of the underlying model (Tab. D.7) and the scores' hyperparameters (Tabs. D.5 and D.6).

## 6 CONCLUSIONS

We introduce a multi-score conformal prediction framework that combines multiple nonconformity scores to select regions in the high-dimensional score space. This approach ensures the desired coverage while minimizing the inclusion of false labels. Unlike existing methods, our technique requires no optimization—neither black-box nor gradient-based—and is applicable to any coverage level. To achieve this, we construct the score using a cost-efficient self-ensemble model with multiple classification heads, trained with both CE and diversity losses. Experimental results demonstrate that our multi-score framework outperforms state-of-the-art baselines across multiple benchmarks.

ACKNOWLEDGMENTS

This work was supported by the Israeli Ministry of Innovation, Science and Technology (Grant No. 1001818518).

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

# A  MATHEMATICAL DETAILS

## A.1  NONCONFORMITY SCORES

A commonly used nonconformity score is based on the confidence of the predicted probabilities, defined as $s_{\text{Thr}}(x, y) = 1 - \pi(x)_y$ (Sadinle et al., 2019). However the Thr score can sometimes undercover hard examples and overcover trivial ones. Hence, a popular alternative is the adaptive prediction sets (APS) method (Romano et al., 2020), which is based on the cumulative probability $s_{\text{APS}}(x, y) := \sum_{q=1}^{Q} \pi(x)_q \mathbf{1} \left[ \pi(x)_q > \pi(x)_y \right] + u \cdot \pi(x)_y$, where $u$ a uniform random value that breaks potential ties between different scores. However, the APS method often results in large prediction sets, which is undesirable. To address this, RAPS score was introduced (Angelopoulos et al., 2020), which encourages smaller prediction sets by penalizing less likely labels. The RAPS score is defined as:

$$s_{\text{RAPS}}(x, y) := \sum_{q=1}^{Q} \pi(x)_q \mathbf{1} \left[ \pi(x)_q > \pi(x)_y \right] + u \cdot \pi(x)_y + \nu \cdot \max(o(x, y) - \kappa, 0), \qquad (14)$$

where $o(x, y)$ denotes the rank of class $y$, and $\nu$ and $\kappa$ are hyperparameters that control the penalty strength. More recently, alternative scoring methods have been proposed that rely on the relative rank of the prediction (Huang et al., 2024; Luo & Zhou, 2024a). For example, the SAPS score is defined as (Huang et al., 2024):

$$s_{\text{SAPS}}(x, y) := \begin{cases} u \cdot \pi_{\max}(x), & \text{if } o(x, y) = 1, \\ \pi_{\max}(x) + (o(x, y) - 2 + u) \cdot \xi, & \text{otherwise,} \end{cases} \qquad (15)$$

where $\xi$ is a hyperparameter that controls the weight of the ranking information, and $\pi_{\max}(x)$ is the maximum softmax probability.

## A.2  TOY EXAMPLE DETAILS AND ADDITIONAL JUSTIFICATION FOR MULTI-DIMENSIONAL CONFORMAL PREDICTION

Assume a binary classification problem with $Y \in \{-1, 1\}$ and prior probabilities $\mathbb{P}(Y = 1)$ and $\mathbb{P}(Y = -1)$. The input $X$ is generated from a mixture of two Gaussians, with $\mathbb{P}(X|Y) = \mathcal{N}(Y, \sigma_y^2)$. In this example, we can compute the posterior $\mathbb{P}(Y|X)$ using Bayes rule:

$$\mathbb{P}(Y = a|X) = \frac{\mathbb{P}(Y = a)\mathbb{P}(X|Y = a)}{\mathbb{P}(Y = 1)\mathbb{P}(X|Y = 1) + \mathbb{P}(Y = -1)\mathbb{P}(X|Y = -1)}, \ \ a = \{-1, 1\}. \qquad (16)$$

We set $\mathbb{P}(Y = 1) = \mathbb{P}(Y = -1) = 0.5$, $\sigma_1^2 = \sigma_{-1}^2 = 0.75$. Let $\mathbb{P}(x, y) = \mathbb{P}(Y = y|X = x)$, consider the following three classifiers:

$$\pi_0(x) = \mathbb{P}(y|x), \ \pi_1(x) := \begin{cases} \mathbb{P}(y|x), & \text{if } x \le 0, \\ (\epsilon, 1 - \epsilon), & \text{if } x > 0 \end{cases}, \ \pi_2(x) := \begin{cases} (\epsilon, 1 - \epsilon), & \text{if } x \le 0, \\ \mathbb{P}(y|x), & \text{if } x > 0 \end{cases} \qquad (17)$$

where $\epsilon \sim \mathcal{N}(0, 1)$. The classifiers are illustrated in Fig. A.1 . Here $\pi_0(x)$ represents the ideal classifier, and $\pi_1(x)$ and $\pi_2(x)$ are ideal only in half of the range of $x$ and uninformative for the other half. We generate 2000 points based on $\mathbb{P}(X|Y)$, and use 1000 for validation and 1000 for calibration. We compute the Thr nonconformity score. Results are averaged over 20 random trials.

When $\mathbb{P}(y|x) = \mathbb{P}(Y = y|X = x)$ is known, it was shown that the optimal set with minimal size under coverage constraint is given by $\Gamma^*(x) = \{y \in \mathcal{Y} | \mathbb{P}(x, y) > q_\alpha\}$ (Lei & Wasserman,

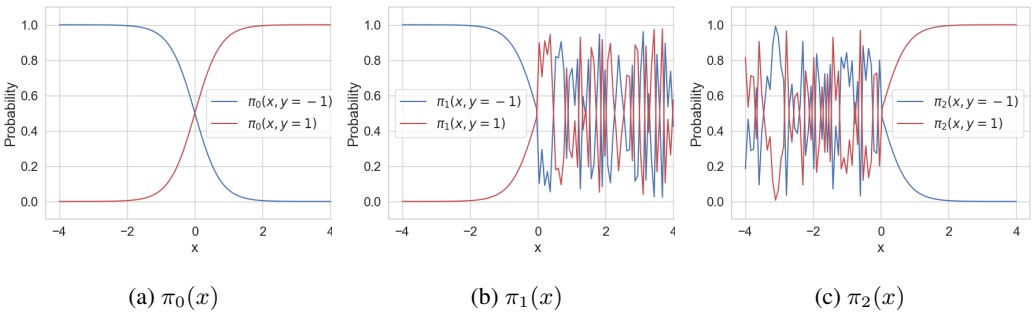

(a) $\pi_0(x)$        (b) $\pi_1(x)$        (c) $\pi_2(x)$

Figure A.1: The three classifiers in the toy example binary classification problem.

2014; Sadinle et al., 2019; Kiyani et al., 2024). This implies that the optimal set is a level set of the distribution $\mathbb{P}(x, y)$. Thus, in our toy example, thresholding $s_0(x, y) = 1 - \mathbb{P}(x, y)$ results in the optimal set. Using only $s_1(x, y)$ and $s_2(x, y)$ the optimal set can be equivalently defined as:

$$\Gamma^*(x) = \{y \in \mathcal{Y} | (\mathbf{1}\{x \le 0\} \cdot s_1(x, y) + \mathbf{1}\{x > 0\} \cdot s_2(x, y)) < 1 - q_\alpha\}$$
$$= \{y \in \mathcal{Y} | \mathbf{s}^T(x, y) \cdot \mathbf{i}_x(x) < 1 - q_\alpha\}$$

where $\mathbf{i}_x(x) = [\mathbf{1}\{x \le 0\}, \mathbf{1}\{x > 0\}]^\top$. Thus, we obtain that the optimal set is a function of the 2-dimensional nonconformity score $\mathbf{s}(x, y)$. In this example, each classifier specializes on a different subdomain of the input space $\mathcal{X}$.

Another practical case is when classifiers specialize on different parts of the output space $\mathcal{Y}$. For example, consider $\mathcal{Y} = \{0, 1, 2\}$, and the following three classifiers:

$$\pi_a(x) := \begin{cases} \mathbb{P}(y = a | x), & \text{if } y = a, \\ \epsilon, & \text{if } y = (a + 1) \bmod 3, \quad a \in \{0, 1, 2\}, \\ 1 - \mathbb{P}(y = a | x) - \epsilon, & \text{if } y = (a + 2) \bmod 3 \end{cases} \qquad (18)$$

where $\epsilon \sim \mathcal{N}(0, 1)$. In this case, the optimal set is given by:

$$\Gamma^*(x) = \{y \in \mathcal{Y} | \mathbf{s}^T(x, y) \cdot \mathbf{i}_y(y) < 1 - q_\alpha\} \qquad (19)$$

where $\mathbf{i}_y(y) = [\mathbf{1}\{y = 1\}, \mathbf{1}\{y = 2\}, \mathbf{1}\{y = 3\}]^\top$. We conclude that whenever $\mathbb{P}(x, y) = \phi(\mathbf{s}(x, y); x, y)$, where $\phi : \mathcal{S} \times \mathcal{X} \times \mathcal{Y} \to [0, 1]$ is a non-degenerate function of the multi-score vector, the optimal set relies on $\mathbf{s}(x, y)$, i.e.:

$$\Gamma^*(x) = \{y \in \mathcal{Y} | \phi(\mathbf{s}(x, y); x, y) < 1 - q_\alpha\}. \qquad (20)$$

In contrast, relying solely on a single score will result in a suboptimal solution. Note that, according to Eq. (20), the ideal set corresponds to a level set of $\phi(\mathbf{s}(x, y); x, y)$, rather than $\mathbf{s}(x, y)$ itself. This implies that, in general, the decision boundaries in the multi-dimensional score space can be arbitrarily complex, depending on the properties of $\phi$.

We conclude that in practical scenarios, where a single score does not provide the full information on the conditional distribution $\mathbb{P}(y|x)$, we benefit from using a multi-dimensional score $\mathbf{s}(x, y)$. It may appear that optimizing for set size efficiency in the multi-score space exponentially increases the number of possible prediction sets to be considered, which makes the optimization more challenging compared to the single-dimensional case. However, our cell partitioning and ranking procedure relaxes the problem to a convenient structured prediction with a simple selection rule that does not require any iterative optimization procedures. Note that the number of cell centers and the summation operation over all scores that fall in the chosen region, remain fixed regardless of the dimensionality of $\mathbf{s}(x, y)$. However, as $n$ increases the cells move apart from each other, when the scores are nonidentical and provide complementary information. Moreover, if each dimension contributes information about the actual conditional distribution $\mathbb{P}(Y|X)$, we anticipate an improved separation between true and false scores. Consequently, the selected subset of cells is expected to exhibit lower $D_i$ values, leading to smaller prediction sets. This is demonstrated in Fig. A.2 presenting the distribution of $D_i$ for the chosen cells. We observe that as $n$ increases, the values of $D_i$ become smaller.

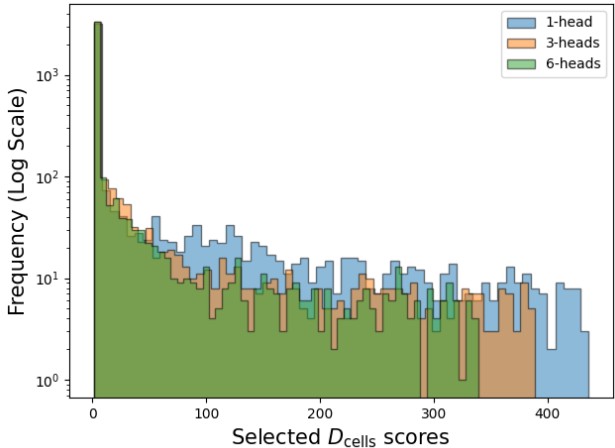

Figure A.2: Histogram of cell scores $D_i$ for the selected number of cells, comparing different number of heads. Values correspond to CIFAR100 dataset, RAPS scores and $\alpha = 0.1$

### A.3 PROOF OF PROPOSITION (2)

*Proof.* For a pair of test input $X_{m+1}$ and a (candidate) label $y \in \mathcal{Y}$ we compute the multidimensional score $\mathbf{s}(X_{m+1}, y) = [s_1(X_{m+1}, y), \dots, s_n(X_{m+1}, q)]^T$. We define a combined score function $s_{\text{multi}} : \mathcal{X} \times \mathcal{Y} \to \mathbb{R}$ that maps the multi-dimensional score $\mathbf{s}(x, y)$ into a single-dimensional score $s_{\text{multi}}(x, y)$. The mapping is defined as follows:

$$s_{\text{multi}}(X_{m+1}, y) = \underset{1 \leq j \leq k'}{\arg \min} \left\| \mathbf{s}(X_{m+1}, y) - \mathbf{s}(X_{(j)}, Y_{(j)}) \right\| \tag{21}$$

where $(j)$ denotes the index of the samples in $\mathcal{D}_{\text{cells}}$ after sorting according to the ratio values in Eq. (7) and eliminating repeating elements, i.e. $(X_{(1)}, Y_{(1)}), \dots, (X_{(k')}, Y_{(k')})$ are the centers of the cells $\mathcal{C}_{(1)}, \dots, \mathcal{C}_{(k')}$ where $D_{(1)} \leq D_{(2)} \leq \cdots \leq D_{(k')}$. This way each pair $(X_{m+1}, y)$ is associated with the closet center and the ranking of this cell serves as the one-dimensional score, defined in Eq. (21).

The prediction set $\Gamma_{\eta^*}(X_{m+1})$ is defined in Eq. (9) by including predictions associated with scores that reside in the selected region. This can be equivalently written in terms of a threshold operation over $s_{\text{multi}}$, as we can write:

$$
\begin{aligned}
\Gamma_{\eta^*}(X_{m+1}) &= \left\{ y \in \{1, \dots, Q\} \mid \mathbf{s}(X_{m+1}, y) \in \mathcal{C}_{\text{in}}^{\eta^*} \right\} \\
&= \left\{ y \in \{1, \dots, Q\} \mid \mathbf{s}(X_{m+1}, y) \in \bigcup_{i=1}^{\eta^*} \mathcal{C}_{(i)} \right\} \\
&= \left\{ y \in \{1, \dots, Q\} \mid \bigcup_{i=1}^{\eta^*} \left[ \mathbf{s}(X_{m+1}, y) \in \mathcal{C}_{(i)} \right] \right\} \\
&\overset{(6)}{=} \left\{ y \in \{1, \dots, Q\} \mid \bigcup_{i=1}^{\eta^*} \left[ i = \underset{1 \leq j \leq k'}{\arg \min} \left\| \mathbf{s}(X_{m+1}, y) - \mathbf{s}(X_{(j)}, Y_{(j)}) \right\| \right] \right\} \\
&\overset{(21)}{=} \left\{ y \in \{1, \dots, Q\} \mid \bigcup_{i=1}^{\eta^*} \left[ i = s_{\text{multi}}(X_{m+1}, y) \right] \right\} \\
&= \left\{ y \in \{1, \dots, Q\} \mid s_{\text{multi}}(X_{m+1}, y) \leq \eta^* \right\}.
\end{aligned}
\tag{22}
$$

This formulation is the same as a standard one-dimensional conformal prediction procedure, where we define the prediction set by thresholding a one-dimensional score function. The threshold $\eta^*$ was

chosen to obtain exact coverage over the held-out data $\mathcal{D}_{\text{re-cal}}$, therefore:

$$\eta^* = \min\{\eta \in \{1, \ldots, k'\} \mid Y_i \in \mathcal{C}_{\text{in}}^\eta \text{ for at least } \lceil(1-\alpha)(r+1)\rceil \text{ samples } (X_i, Y_i) \in \mathcal{D}_{\text{re-cal}}\}$$

$$\stackrel{(22)}{=} \min\{\eta \in \{1, \ldots, k'\} \mid s_{\text{multi}}(X_i, Y_i) \leq \eta \text{ for at least } \lceil(1-\alpha)(r+1)\rceil \text{ samples in } \mathcal{D}_{\text{re-cal}}\}$$

$$= \text{Quantile}\left(\frac{\lceil(r+1)(1-\alpha)\rceil}{r}; \{s_{\text{multi}}(X_i, Y_i)\}_{i=k+1}^m\right).$$

Thus, assuming that $\mathcal{D}_{\text{re-cal}}$ and $(X_{m+1}, Y_{m+1})$ are exchangeable , we obtain that:

$$\mathbb{P}\left(Y_{m+1} \in \Gamma_{\eta^*}(X_{m+1})\right) \geq 1 - \alpha \tag{23}$$

following Theorem (1). $\qquad\square$

Note that the split of $\mathcal{D}_{\text{cal}}$ to two disjoint subsets $\mathcal{D}_{\text{cells}}$ and $\mathcal{D}_{\text{re-cal}}$ is critical here, since we cannot claim that the samples in $\mathcal{D}_{\text{cells}}$ and the test pair $(X_{m+1}, Y_{m+1})$ are exchangable as $\mathcal{D}_{\text{cells}}$ serves for the computation of the score defined in (21) (in a similar way to the fact that the training data cannot be used for calibration in standard conformal prediction).

## A.4 EQUIVALENCE BETWEEN CELL SELECTION AND SET-SIZE OPTIMIZATION

We start by proving Lemma 3.

*Proof.* Let $F_i = \sum_{j=1}^k \sum_{q=1}^Q \mathbf{1}\{s(X_j, q) \in \mathcal{C}_i\} \cdot \mathbf{1}\{Y_j \neq q\}$ and $T_i = \sum_{j=1}^k \mathbf{1}\{s(X_j, Y_j) \in \mathcal{C}_i\}$ denote the number of scores with false labels and true labels, respectively. Let $I' \subseteq 2^{k'}$ denote the set of disjoint cell indexes, removing duplicates. Using these definitions, a finite-sample approximation of the expected set-size $L_{\text{size}}$ over $\mathcal{D}_{\text{cells}}$, can be written as:

$$\hat{L}_{\text{size}} = \frac{1}{k} \sum_{j=1}^k \sum_{q=1}^Q \mathbf{1}\{q \in \Gamma^I(X_j)\}$$

$$\stackrel{(9)}{=} \frac{1}{k} \sum_{j=1}^k \sum_{q=1}^Q \mathbf{1}\{S(X_j, q) \in \cup_{i \in I}\mathcal{C}_i\}$$

$$= \frac{1}{k} \sum_{j=1}^k \sum_{q=1}^Q \mathbf{1}\{S(X_j, q) \in \cup_{i \in I'}\mathcal{C}_i\}$$

$$= \frac{1}{k} \sum_{j=1}^k \sum_{i \in I'} \sum_{q=1}^Q \mathbf{1}\{s(X_j, q) \in \mathcal{C}_i\}$$

$$= \frac{1}{k} \sum_{i \in I'} F_i + T_i$$

$$= \frac{1}{k} \sum_{i \in I} \frac{F_i + T_i}{T_i}$$

$$= \frac{1}{k} \sum_{i \in I} D_i, \tag{24}$$

where the final equality directly follows from the definition of the cell score in Eq. (7). For the coverage constraint, we obtain:

$$\hat{L}_{\text{coverage}} = \frac{1}{k} \sum_{j=1}^k \mathbf{1}\{Y_j \in \Gamma^I(X_j)\} = \frac{1}{k} \sum_{i \in I'} T_i = \frac{1}{k} \sum_{i \in I} 1 = \frac{|I|}{k} \tag{25}$$

$\qquad\square$

The proof of Proposition 4 follows directly, as we now state.

---

**Algorithm B.1** Jackknife+ Multi-Score Conformal prediction

---

**Definitions:** $\mathbf{s}(x, y)$ is a multi-dimensional score function. $\mathcal{D}_{\text{cal}}$ is the calibration data of size $m$. $X_{m+1}$ is a new test sample, $\alpha$ is the miscoverage level, $k$ is the number of samples for cell-partitioning and scoring, and $r = m - k$ is the number of samples for re-calibration.

1: **function** MULTI-SCORE-CP($\boldsymbol{s}(x, y), \mathcal{D}_{\text{cal}}, \alpha$)
2:      Compute the scores $\mathbf{s}(X_i, Y_i), \ i \in \{1, \ldots, m\}$
3:      Segment $\mathcal{S}$ into cells $\{\mathcal{C}_i\}_{i=1}^{k}$ centered at $\{(X_i, Y_i)\}_{i=1}^{k}$
4:      Remove duplicate cells $\mathcal{C}_1, \mathcal{C}_1, \ldots, \mathcal{C}_{k'}$
5:      **for** $i \in \{1, \ldots, k\}$ **do**
6:          Compute $D_j^{-i}, j = 1, \ldots, i-1, i+1, \ldots, k'$ using Eq. (7), excluding the $i$-th point
7:          Rank the cells $\mathcal{C}_{(1)}, \mathcal{C}_{(2)}, \ldots, \mathcal{C}_{(k'-1)}$ according to $D_{(1)}^{-i} \leq D_{(2)}^{-i} \leq \ldots \leq D_{(k'-1)}^{-i}$
8:          $E_i^{-i} \leftarrow \arg\min_{j \in \{1, \ldots, k'-1\}} \left\| \mathbf{s}(X_i, Y_i) - \mathbf{s}(X_{(j)}, Y_{(j)}) \right\|$
9:          $E_{m+1}^{-i}(y) \leftarrow \arg\min_{j \in \{1, \ldots, k'-1\}} \left\| \mathbf{s}(X_{m+1}, y) - \mathbf{s}(X_{(j)}, Y_{(j)}) \right\|, \ y \in \mathcal{Y}$
10:      $\Gamma_{JK+}(X_{m+1}) = \left\{ y \in \{1, \ldots, Q\} \, | \, \sum_{i=1}^{m} \mathbf{1}\{E_i^{-i} < E_{m+1}^{-i}(y)\} < (1-\alpha)(m+1) \right\}$
11:      **return** $\Gamma_{JK+}(X_{m+1})$

---

*Proof.* Lets first assume that there are no duplicates, i.e. $T_i = 1, \forall 1 \leq i \leq k$. Let $I^*$ denote the $(1 - \alpha)$ proportion of cells with the smallest $D_i$ values, we obtain that:

$$\hat{L}_{\text{coverage}}^* = \frac{|I^*|}{k} = 1 - \alpha. \tag{26}$$

By construction, $I^*$ has the smallest empirical set size compared to any other set $I^\alpha$ with $|I^\alpha| = |I^*| = 1 - \alpha$:

$$\hat{L}_{\text{size}}^* = \frac{1}{k} \sum_{i \in I^*} D_i \leq \frac{1}{k} \sum_{i \in I^\alpha} D_i = \hat{L}_{\text{size}}^\alpha. \tag{27}$$

Thus, when there are no duplicates, the solution to the optimization problem in Eq. (12) is obtained by ordering the cells according to $D_i$ and selecting the top-ranked $1 - \alpha$ proportion of cells. When duplicates exist, it can be interpreted as selecting the same cell $T_i$ times. In this case, the solution to the optimization problem in Eq. (12) is obtained by ordering the cells according to $D_i$ and selecting the top-ranked cells until a $1 - \alpha$ proportion of $k$ is selected, subject to the constraint that all duplicates must be selected together. It is possible for some disjoint cells to have the same $D_i$ scores with different $T_i$ values, which would make it ambiguous to determine which cell to select first. However, this will only make difference when approaching the selection limit and since, anyway, duplicates are rare in dimensions higher than one, this ambiguity has a negligible effect.

Finally, re-calibration over $\mathcal{D}_{\text{re-cal}}$, as defined in Eq. (8), guarantees exact coverage, as follows from Proposition 2.        $\square$

## B    ALGORITHMS

The jackknife+ variant, as described in Section D.6, is provided in Algorithm B.1, while the soft centers variant, as detailed in Section D.6, is presented in Algorithm B.2.

## C    IMPLEMENTATION AND DATASET DETAILS

**Datasets and Implementation.** The details of each dataset and the corresponding data splits are summarized in Table C.1, where Tiny ImageNet, CIFAR100 and PathMNIST are used in our main results while ImageNet and 20 Newsgroups are used for the additional experiments, described in § D. The calibration data is split into half for cell computation, and re-calibration.

---

**Algorithm B.2** Soft Multi-Score Conformal Prediction

---

**Definitions:** $\mathbf{s}(x, y)$ is a multi-dimensional score function. $\mathcal{D}_{\text{cal}}$ is the calibration data of size $m$. $X_{m+1}$ is a new test sample, $\alpha$ is the miscoverage level, $k$ is the number of samples for cell-partitioning and scoring, and $r = m - k$ is the number of samples for re-calibration and b is the number of neighbors.

1: **function** SOFT MULTI-SCORE-CP($\mathbf{s}(x, y)$, $\mathcal{D}_{\text{cal}}$, $\alpha$, b)
2:      Randomly split $\mathcal{D}_{\text{cal}}$ to $\mathcal{D}_{\text{cells}} = \{(X_i, Y_i)\}_{i=1}^{k}$ and $\mathcal{D}_{\text{re-cal}} = \{(X_i, Y_i)\}_{i=k+1}^{m}$
3:      Compute the scores $\mathbf{s}(X_i, Y_i)$, $i \in \{1, \ldots, m\}$
4:      Segment $\mathcal{S}$ into cells $\{\mathcal{C}_i\}_{i=1}^{k}$ centered at $\{(X_i, Y_i)\}_{i=1}^{k}$
5:      Compute $D_i, i = 1, \ldots, k$ according to Eq. (7)
6:      Remove duplicate cells $\mathcal{C}_1, \mathcal{C}_1, \ldots, \mathcal{C}_{k'}$
7:      Rank the cells $\mathcal{C}_{(1)}, \mathcal{C}_{(2)}, \ldots, \mathcal{C}_{(k')}$ according to $D_{(1)} \leq D_{(2)} \leq \ldots \leq D_{(k')}$
8:      $\mathcal{T}^b(i) \leftarrow$ b nearest cells of $\mathbf{s}(X_i, Y_i)$ in $\{\mathcal{C}_{(i)}\}_{i=1}^{k'}$, $i \in \{k+1, \ldots, m\}$
9:      $\eta^* \leftarrow \min\left\{\eta \left| \sum_{i=k+1}^{m} \mathbf{1}\left\{\left(\sum_{t=1}^{b} \mathbf{1}\left\{\mathcal{T}_t^b(i) \subseteq C_{\text{in}}^{\eta}\right\}\right) > \lceil 0.5 \cdot b \rceil\right\} \geq \lceil (1-\alpha)(r+1) \rceil\right.\right\}$
10:      $\mathcal{C}_{\text{in}}^{\eta^*} \leftarrow \bigcup_{i=1}^{\eta^*} \mathcal{C}_{(i)}$
11:      **return** $\mathcal{C}_{\text{in}}^{\eta^*}$
12: **function** SOFT MULTI-SCORE-EVALUATION($\mathbf{s}(x, y)$, $X_{m+1}$, $\mathcal{C}_{\text{in}}^{\eta^*}$)
13:      Compute the scores $\mathbf{s}(X_{m+1}, y)$, $y \in \{1, \ldots, Q\}$
14:      $\mathcal{T}^b(y) \leftarrow$ b nearest cells of $\mathbf{s}(X_{m+1}, y)$ in $\{\mathcal{C}_{(i)}\}_{i=1}^{k'}$, $y \in \mathcal{Y}$
15:      $\Gamma_{\eta^*}(X_{m+1}) = \left\{y \in \mathcal{Y} \left| \left(\sum_{t=1}^{b} \mathbf{1}\left\{\mathcal{T}_t^b(y) \subseteq C_{\text{in}}^{\eta^*}\right\}\right) > \lceil 0.5 \cdot b \rceil\right.\right\}$
16:      **return** $\Gamma_{\eta^*}(X_{m+1})$

---

Table C.1: Datasets Details

| **Dataset** | # Classes | Train | Validation | Calibration | Test | Average Accuracy |
|---|---|---|---|---|---|---|
| Tiny ImageNet | 200 | 71,500 | 11,000 | 16,500 | 11,000 | 0.58 |
| CIFAR100 | 100 | 39,000 | 6,000 | 9,000 | 6,000 | 0.69 |
| PathMNIST | 9 | 69,667 | 10,718 | 16,077 | 10,718 | 0.94 |
| 20 Newsgroups | 20 | 9,800 | 2,449 | 3,298 | 3,299 | 0.87 |
| ImageNet | 1,000 | 1,281,184 | 10,000 | 20,000 | 20,000 | 0.71 |

For the first three datasets we used ResNet50 model with pretrained weights on ImageNet. Each head is a 3 layer feed-forward neural network with, BatchNorm, ReLU activation and dropout with $p = 0.1$.

In the first stage, the full model with a single classification head was fine-tuned on each task with 20, 100 and 200 epochs for Tiny ImageNet, CIFAR100 and PathMNIST, respectively. In the second stage, we freeze the backbone model and train only the classification heads for 20 epochs, using the loss defined in Eq. (13). In both stages, we use Adam optimizer with cosine annealing scheduler, momentum decay of $0.95$, weight decay of $1e - 5$, and batch size of 16.

For the computation of the RAPS score we used $\lambda = 0.05$ and $\kappa = 5$, and for the SAPS score we set $\xi = 0.3$. Performance with respect to other parameter values is reported in Tables D.5 and D.6 for RAPS and SAPS, respectively.

## D    ADDITIONAL RESULTS

### D.1    EVALUATING OTHER PERFORMANCE ASPECTS

**Conditional Coverage.** We assess performance in terms of conditional coverage, evaluated using size-stratified coverage violation (SSCV) and worst-slice coverage (WSC). For SSCV, we define a set of disjoint strata $\{S_j\}_{j=1}^{J}$, where $\cup_{j=1}^{J} S_j = \{1, \ldots, Q\}$. We partition the data into groups with

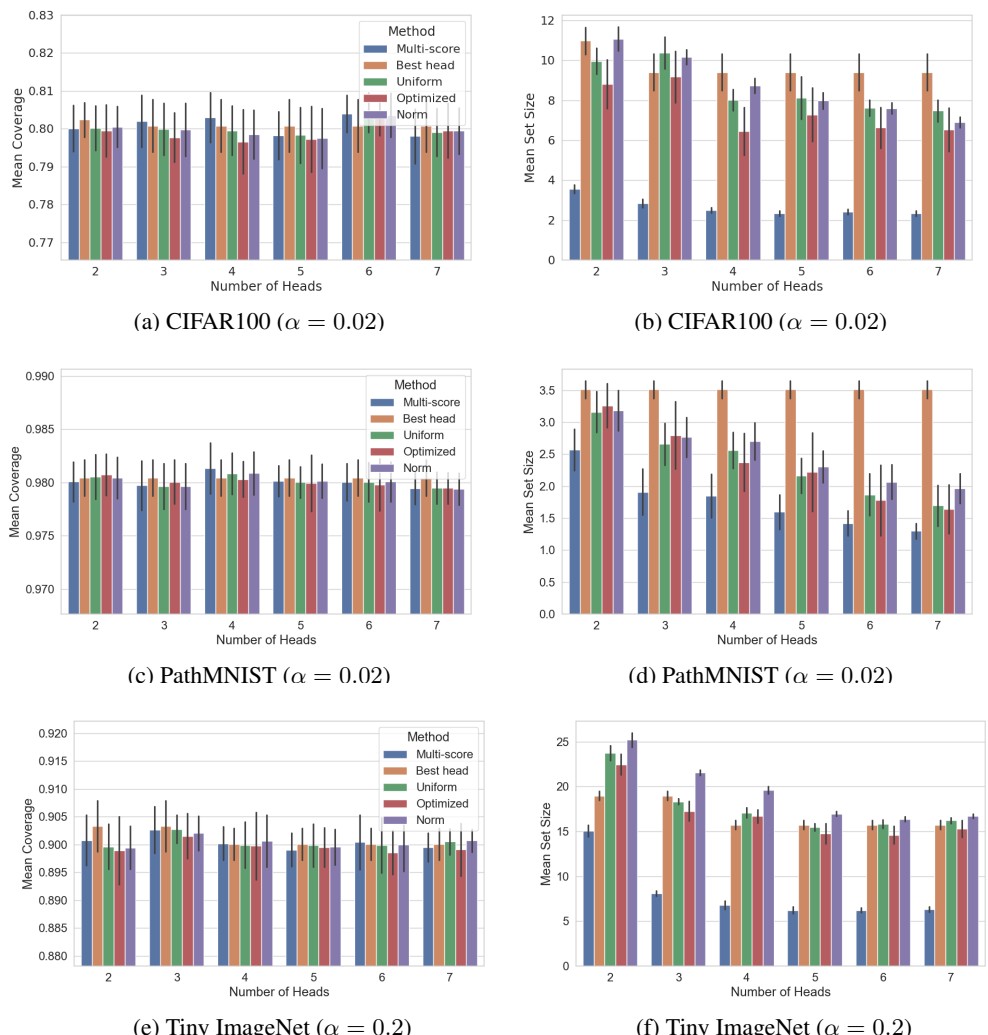

Figure D.1: Conformal prediction with RAPS as a function of the number of classification heads. Results compare multi-score conformal prediction and the baselines (Best head, Optimized, Uniform, and Norm) across two metrics: empirical coverage (left column), and mean set size (right column), over: CIFAR100, Tiny ImageNet and PathMNIST.

equal numbers of samples. Let $q_j = \text{Quantile}\left(\frac{j}{J}; \{|\Gamma(X_i, Y_i)|\}_i\right)$, $j = 0, \ldots, J$, denote the $\frac{j}{J}$-th quantile of the set sizes. The $j$th group is then defined as $\mathcal{G}_j = \{i : q_{j-1} \leq |\Gamma(X_i, Y_i)| \leq q_j\}$ for $j \in \{1, \ldots, J\}$. Accordingly, the SSCV is defined as (Angelopoulos et al., 2020):

$$\text{SSCV}(\{S_j\}_{j=1}^J) = \sup_j \left| \frac{|\{i : Y_i \in \Gamma(X_i, Y_i), i \in \mathcal{G}_j\}|}{|\mathcal{G}_j|} - (1 - \alpha) \right| \tag{28}$$

For this evaluation, we divided the data into $J = 10$ groups. The results, summarized in Table D.1, indicate that all methods achieve similar SSCV scores, with no method showing a clear advantage over the others.

Additionally, we evaluate the WSC, as defined in (Romano et al., 2020), which quantifies the worst-case coverage along a random projection within a local region of the distribution. The results, summarized in Table D.2, indicate that the multi-score method performs comparably to the baselines in terms of WSC, confirming that the proposed approach maintains conditional coverage.

**Results with SAPS scores.** To examine the robustness of the multi-score method with respect to the choice of nonconformity scores, we compare the results achieved using the SAPS nonconformity score across different levels of $\alpha$ and datasets. The results are presented in figs D.2 and D.3. As

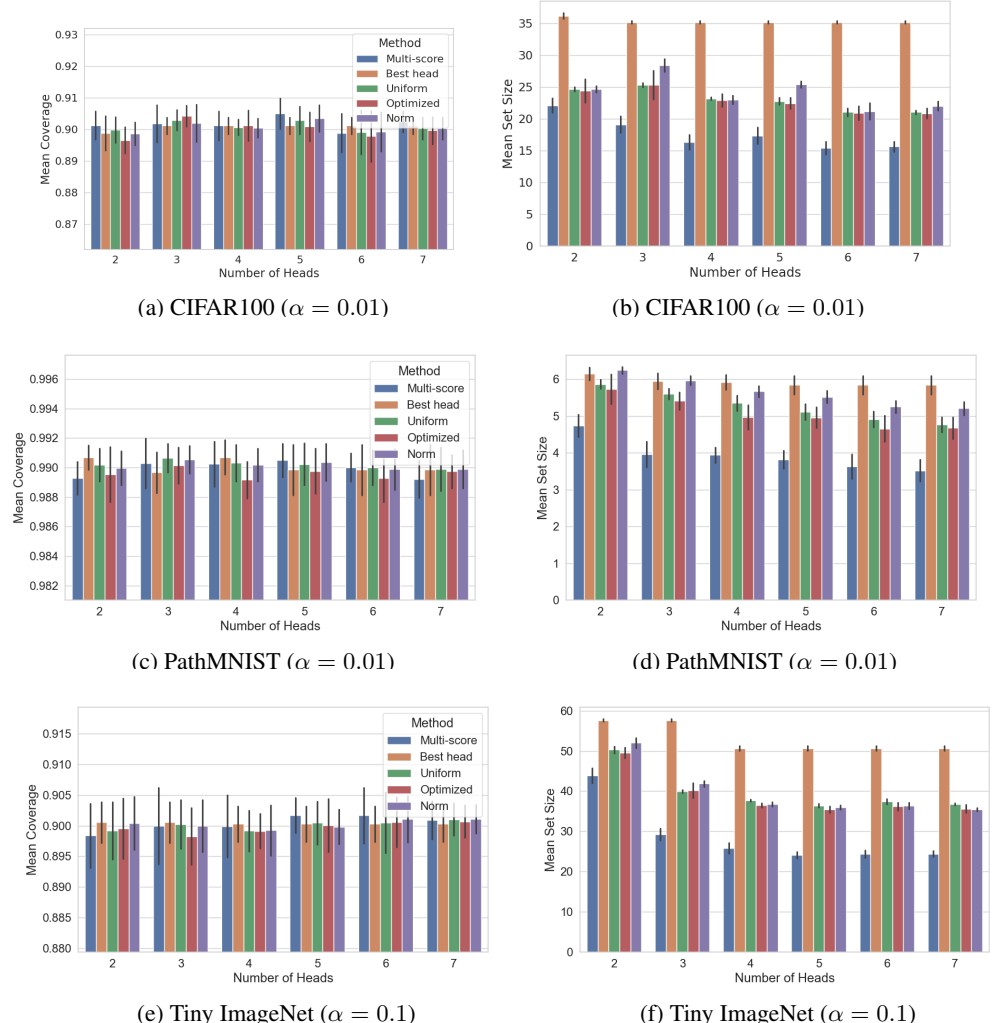

Figure D.2: Conformal prediction with SAPS as a function of the number of classification heads. Results compare multi-score conformal prediction and the baselines (Best head, Optimized, Uniform, and Norm) across two metrics: empirical coverage (left column), and mean set size (right column), over: CIFAR100, Tiny ImageNet and PathMNIST.

expected, all methods achieve the required coverage. The proposed multi-score calibration produces smaller prediction sets, with significant improvements as the number of heads increases.

**Results with Thr and APS scores.** In general, our proposed method can be applied over any type of score. However, we have seen that it is not fully optimal for Thr and APS. The reason is that for these scores the values are more concentrated on specific levels, while for SAPS and RAPS the values are more spread. In order to improve this behavior we use temperature scaling, i.e. we divide the logits by $T$ before applying Softmax(). As $T$ increases the entropy of the probabilities increases and they become more spread, as illustrated in Fig. D.5. Figure D.6 presents the set sizes obtained for all methods with respect to different temperatures. We see that the efficiency the proposed method greatly improves as $T$ increases for both Thr and APS, while the performance of the baseline methods is less effected by $T$. For RAPS and SAPS the proposed method outperforms the baselines regardless of the temperature due to the inherent spread of these scores.

**The distribution of set sizes.** Figure D.7 illustrates how increasing the number of heads (from 1 to 7) shifts the set size distribution toward smaller values, indicating more samples with smaller set sizes. Notably, the multi-score method responds more effectively to the increase in heads compared

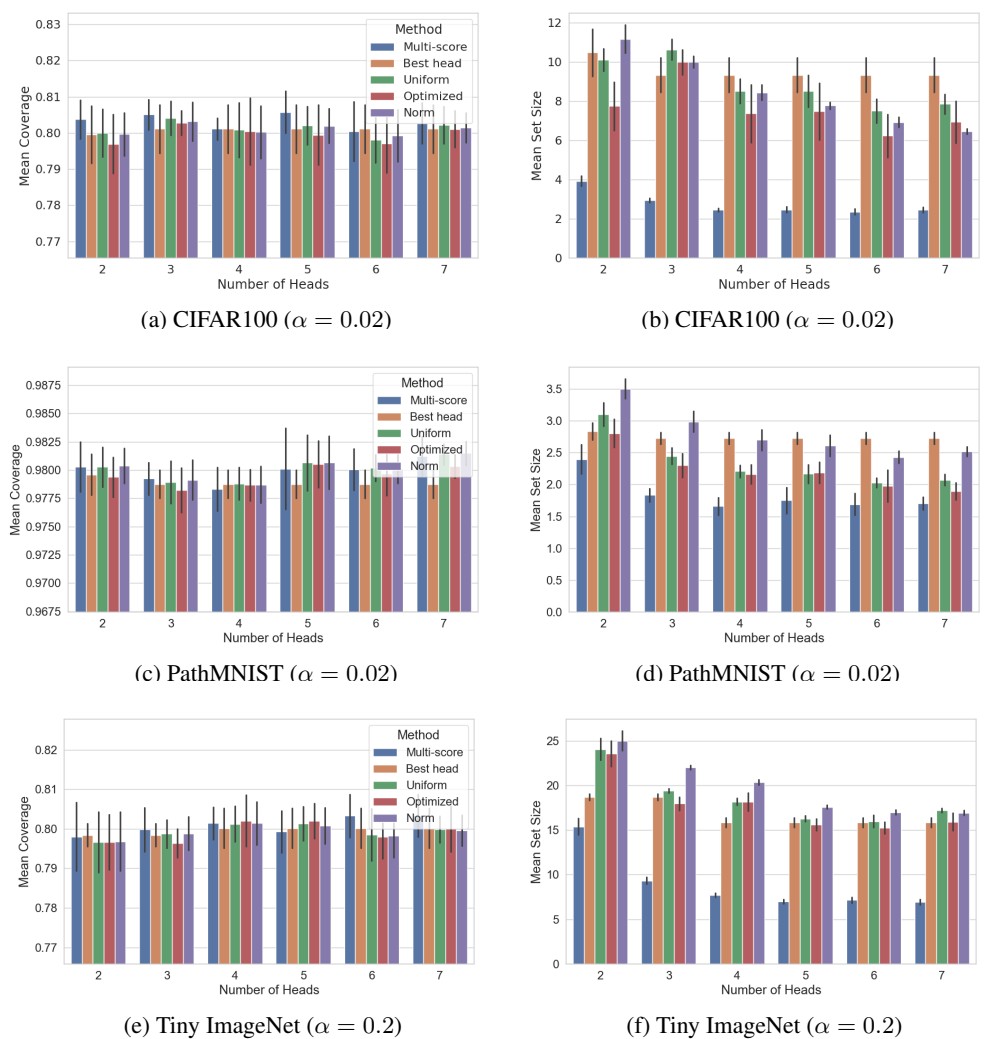

Figure D.3: Conformal prediction with SAPS as a function of the number of classification heads. Results compare multi-score conformal prediction and the baselines (Best head, Optimized, Uniform, and Norm) across two metrics: empirical coverage (left column), and mean set size (right column), over: CIFAR100, Tiny ImageNet and PathMNIST.

to the baselines, achieving set size values in a smaller range $(2-41)$, while for the other methods the sizes range from $11-41$. This highlights again that our multi-score method leads to more efficient and precise sets.

## D.2 RESULTS WITH OTHER TYPES OF MULTIPLE SCORES

**Standard Ensemble.** We conduct an experiment with a standard ensemble, consisting of multiple different models that are trained separately on the same dataset. We use ImageNet dataset for evaluation with an ensemble of 5 models pretrained on ImageNet: VGG16, Inception, ResNet50, ResNet152 and DenseNet161. Results are shown in Fig. D.8 for RAPS score and $\alpha = [0.03, 0.05, 0.1]$. Similarly to our main results with self-ensemble, here too the proposed method obtains the smallest prediction set sizes. This indicates that our method can be applied to self-ensemble models as well as regular ensembles.

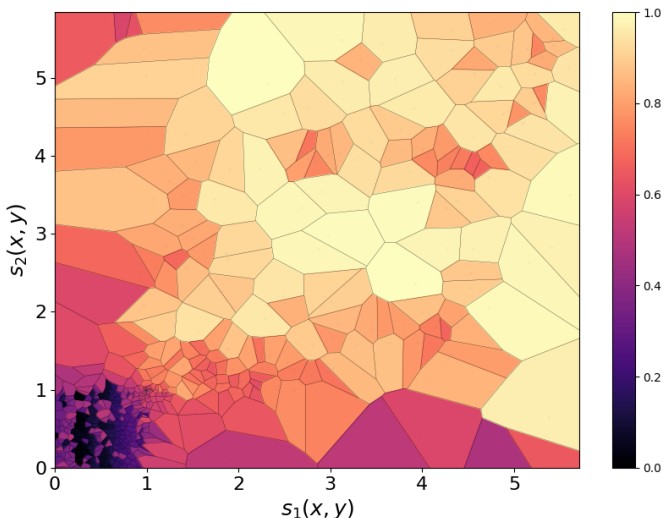

Figure D.4: Visualization of cell selection order over CIFAR100 with RAPS score and $\alpha = 0.1$. Darker cells were selected earlier in the sequence while lighter cells are selected at a later stage.

Table D.1: SSCV measure for CIFAR100 with RAPS score

| $\alpha$ | Head | Methods | | | | |
|---|---|---|---|---|---|---|
| | | Multi-score | Best head | Uniform | Optimized | Norm |
| | 1 | 0.174 | 0.034 | 0.174 | 0.159 | 0.174 |
| | 2 | 0.167 | 0.108 | 0.114 | 0.194 | 0.141 |
| | 3 | 0.240 | 0.104 | 0.066 | 0.162 | 0.277 |
| 0.1 | 4 | 0.298 | 0.204 | 0.178 | 0.167 | 0.264 |
| | 5 | 0.106 | 0.155 | 0.167 | 0.200 | 0.252 |
| | 6 | 0.131 | 0.153 | 0.172 | 0.308 | 0.125 |
| | 7 | 0.056 | 0.159 | 0.062 | 0.191 | 0.175 |
| | 1 | 0.190 | 0.037 | 0.190 | 0.157 | 0.190 |
| | 2 | 0.362 | 0.249 | 0.292 | 0.221 | 0.330 |
| | 3 | 0.342 | 0.200 | 0.132 | 0.187 | 0.098 |
| 0.2 | 4 | 0.379 | 0.206 | 0.167 | 0.166 | 0.279 |
| | 5 | 0.361 | 0.192 | 0.159 | 0.209 | 0.331 |
| | 6 | 0.220 | 0.180 | 0.071 | 0.274 | 0.276 |
| | 7 | 0.186 | 0.203 | 0.200 | 0.171 | 0.245 |

**Test-Time Augmentation.** We evaluate our method on a multi-dimensional score that is formed by test-time augmentations (Lu, 2023). We use ImageNet dataset with Inception-V3 model. We use a simple common test-time augmentation policy (Krizhevsky et al., 2012), which consists of a random crop and a horizontal flip. The random crop pads the original image by 4 pixels and takes a 256x256 crop of the resulting image. We draw five augmentations using this policy. Figure D.9 presents the results for RAPS score and $\alpha = [0.03, 0.05, 0.1]$. Our method outperforms the baselines in terms of prediction set size. We conclude that test-time augmentation can serve as a possible alternative for generating multiple nonconformity scores within our multi-score conformal prediction framework.

**Multiple scores computed over a single head.** We examined a setting where instead of considering multiple classification heads, we use a single head and compute different conformity scores: Thr,

Table D.2: WSC Results on CIFAR100 and TinyImageNet (10 trials)

| $\alpha$ | Dataset | Score | Method | | | | |
|---|---|---|---|---|---|---|---|
| | | | Multi-score | Best head | Uniform | Optimized | Norm |
| 0.1 | CIFAR100 | RAPS | 0.901 | 0.905 | 0.903 | 0.905 | 0.901 |
| | | SAPS | 0.903 | 0.908 | 0.906 | 0.895 | 0.894 |
| | TinyImageNet | RAPS | 0.905 | 0.903 | 0.896 | 0.896 | 0.892 |
| | | SAPS | 0.900 | 0.902 | 0.900 | 0.901 | 0.909 |
| 0.2 | CIFAR100 | RAPS | 0.801 | 0.801 | 0.797 | 0.799 | 0.803 |
| | | SAPS | 0.812 | 0.819 | 0.800 | 0.813 | 0.817 |
| | TinyImageNet | RAPS | 0.796 | 0.791 | 0.801 | 0.800 | 0.802 |
| | | SAPS | 0.808 | 0.802 | 0.808 | 0.806 | 0.807 |

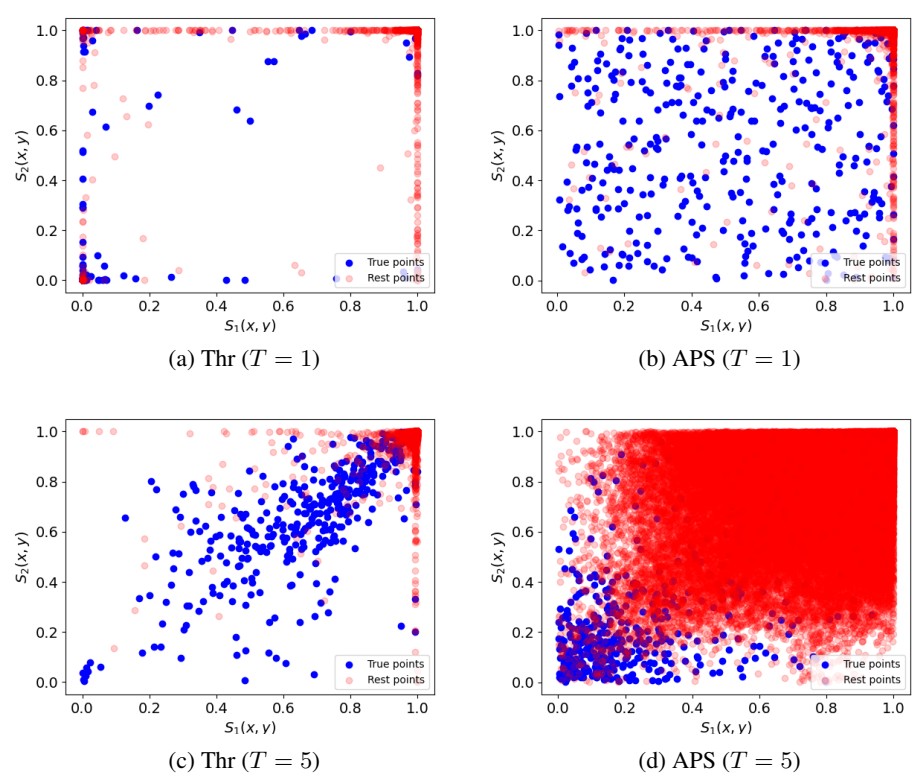

(a) Thr $(T = 1)$

(b) APS $(T = 1)$

(c) Thr $(T = 5)$

(d) APS $(T = 5)$

Figure D.5: Demonstration of APS and Thrt scores's distribution with different Temperatures on CIFAR-100 and $\alpha = 0.1$.

APS, RAPS and SAPS. To ensure all scores are comparable and fall in the range between $[0, 1]$, we apply Softmax() over each score. Table D.3 summarizes the results for all datasets. We see that combining multiple scores improves the results compared to the best single score, and that the proposed method obtains the smallest prediction sets in almost all cases. The advantage of this score fusion is that it does not require any additional modifications to the model or further fine-tuning iterations.

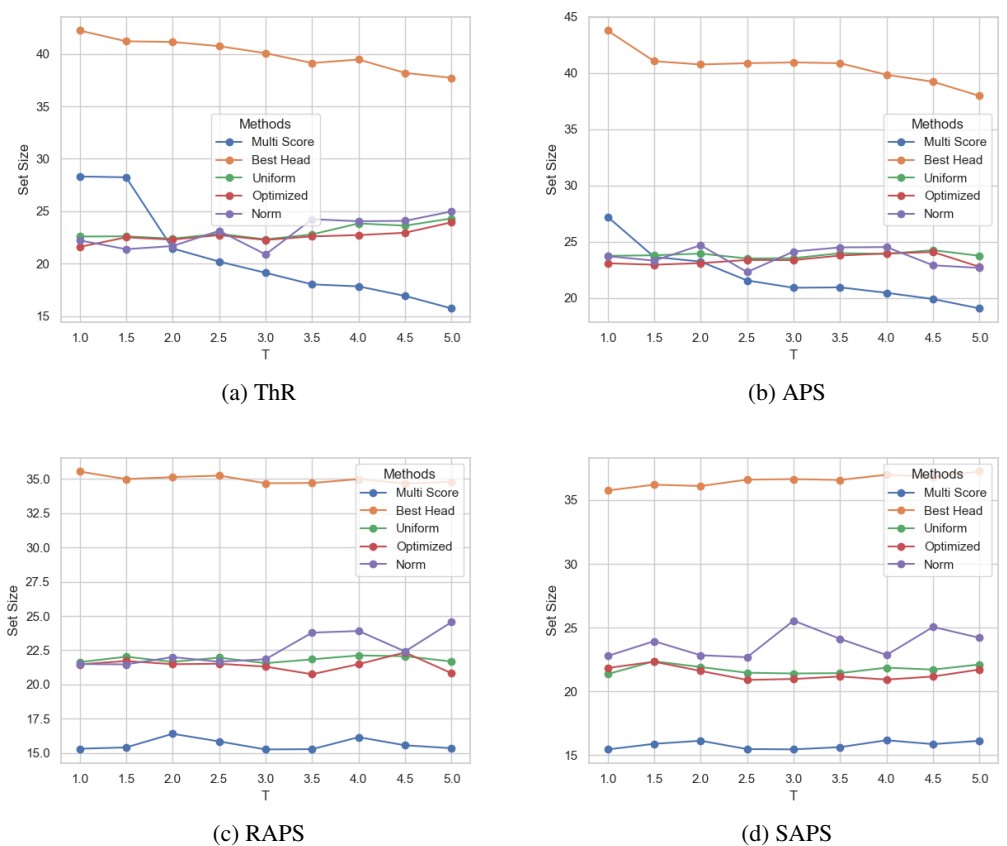

Figure D.6: Set size as a function of the temperature for different nonconformity scores. Results are shown for CIFAR100 with 7 heads and $\alpha = 0.1$

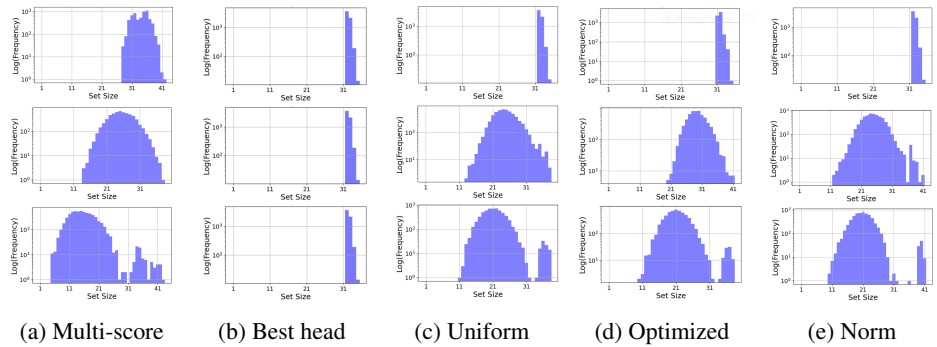

Figure D.7: Histograms of the set size distribution across different numbers of classification heads with RAPS score on Cifar100 and $\alpha$=0.1. Rows represent 1, 2, and 7 heads, while columns correspond to different methods.

## D.3 ROBUSTNESS TO HYPERPARAMETERS

**Influence of training with diversity regularization.** We conducted an ablation study to examine the affect of adding diversity regularization to the loss used for training the classification heads, defined in Eq. (13). We use the same fine-tuned model in the first training stage and change only

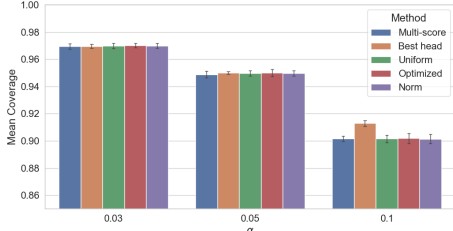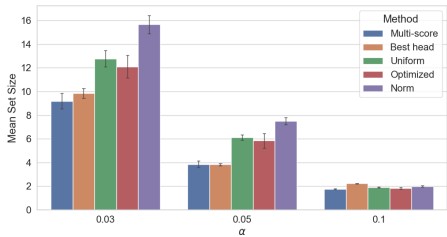

Figure D.8: Results for ImageNet with a standard ensemble of five different models, using RAPS score.

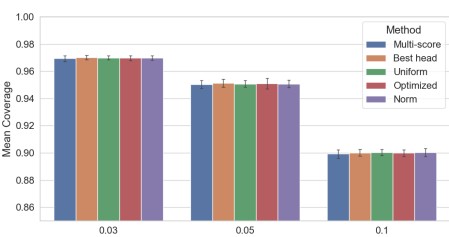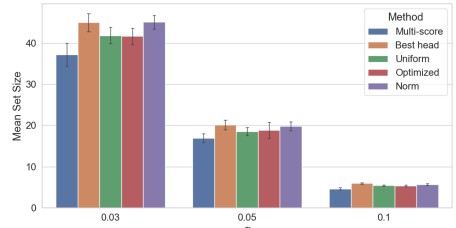

Figure D.9: Results for ImageNet with test-time augmentations, using RAPS score.

Table D.3: Set-size comparison for a single classification head and multiple types of scores.

| $\alpha$ | Dataset | Method | | | | |
|---|---|---|---|---|---|---|
| | | Multi-score | Best head | Uniform | Optimized | Norm |
| 0.1 | CIFAR100 | 33.90 | 35.12 | 35.32 | 34.96 | 35.54 |
| | TinyImageNet | 58.08 | 61.62 | 62.96 | 62.70 | 78.45 |
| 0.15 | CIFAR100 | 18.33 | 27.88 | 28.12 | 27.52 | 28.18 |
| | TinyImageNet | 35.84 | 48.55 | 49.47 | 49.48 | 55.75 |
| 0.2 | CIFAR100 | 8.25 | 9.34 | 9.19 | 8.35 | 13.02 |
| | TinyImageNet | 19.99 | 24.94 | 27.49 | 26.94 | 34.49 |
| 0.01 | PathMNIST | 6.26 | 6.69 | 6.67 | 6.5 | 6.56 |
| 0.02 | PathMNIST | 3.31 | 4.56 | 2.98 | 2.96 | 3.07 |

the second stage to optimize the regularized loss in Eq. (13) with different values of $\lambda$, controlling the strength of the diversity regularization. Figure D.10 shows the similarity between heads with and without the diversity regularization. As expected, the similarity between heads is smaller for the model trained with the regularized loss. Figure D.11 compares the set sizes obtained for different values of $\lambda$, demonstrating that adding the regularization results in smaller sets. The optimal value of $\lambda$ is around $0.8$, after which an increase in set size is observed, apparently due to the fact that increasing head diversity comes at the cost of decreasing the accuracy of each head.

**Influence of the of size** $\mathcal{D}_{\mathbf{cal}}$. We examine how the performance is affected by the size of the calibration data. Here, we vary the proportion $p$ of samples from $\mathcal{D}_{\text{cal}}$ that are actually used, i.e., we select a subset $\mathcal{D}_{\text{cal}}^{p} \subseteq \mathcal{D}_{\text{cal}}$, where $|\mathcal{D}_{\text{cal}}^{p}| = p \cdot |\mathcal{D}_{\text{cal}}|$. Then, as before, $\mathcal{D}_{\text{cal}}^{p}$ is split into half for cell computation, and re-calibration. Figure D.12 presents the set sizes for different values of $p$. It can be seen that the proposed method is almost always preferable, with its advantage becoming more pronounced as the size of the calibration data increases.

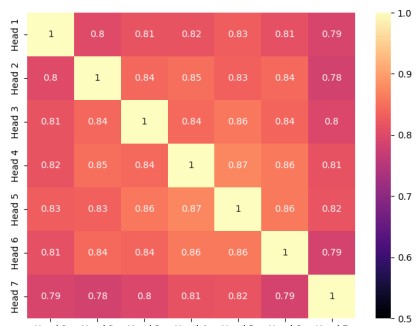 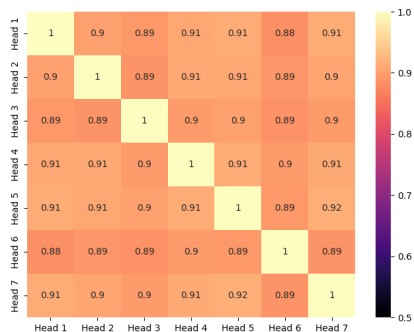

Figure D.10: Similarity matrix between prediction heads for heads trained with (left) or without (right) diversity regularization on CIFAR100 dataset.

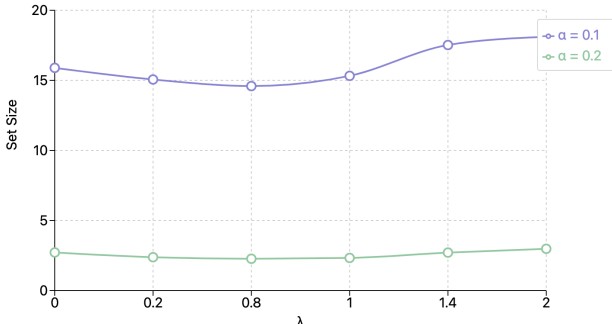

Figure D.11: Set size as a function of the regularization parameter $\lambda$ for CIFAR100 with RAPS score and 7 heads.

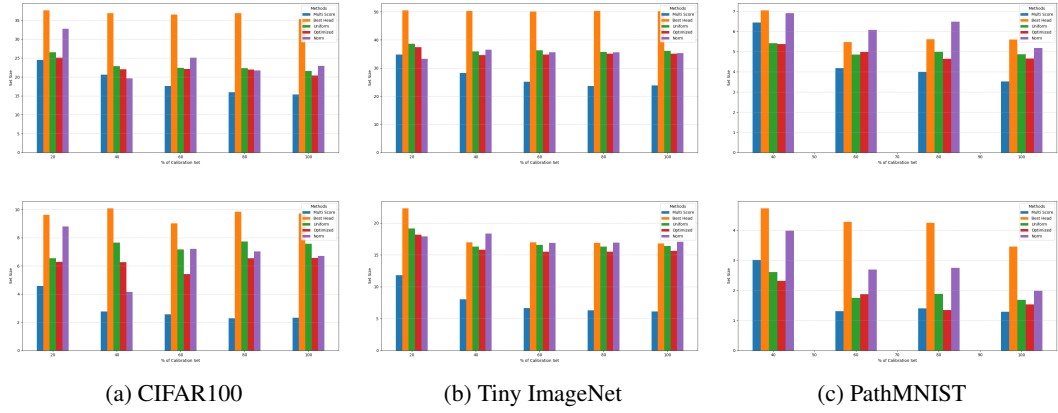

| (a) CIFAR100 | (b) Tiny ImageNet | (c) PathMNIST |

Figure D.12: Sensitivity of the results to the size of $\mathcal{D}_{\text{cal}}$. We select a subset $\mathcal{D}^p_{\text{cal}} \subseteq \mathcal{D}_{\text{cal}}$, where $|\mathcal{D}^p_{\text{cal}}| = p \cdot |\mathcal{D}_{\text{cal}}|$. The set sizes obtained for different values of $p$ are reported for RAPS score on 7 heads. Upper row for $\alpha = 0.1$ and lower row for $\alpha = 0.2$.

**Influence of the of size $\mathcal{D}_{\text{cells}}$.** We investigated the effect of $k$, the size of $\mathcal{D}_{\text{cells}}$, on the results. Recall that $\mathcal{D}_{\text{cells}}$ is responsible to the definition of the cells in Eq. (6) and the computation of the ratio-based scores in Eq. (7). In this experiment, the dataset is divided, as before, into three fixed subsets: $\mathcal{D}_{\text{cells}}$, $\mathcal{D}_{\text{re-cal}}$, and $\mathcal{D}_{\text{test}}$. We vary the proportion $p$ of samples from $\mathcal{D}_{\text{cells}}$ that are actually

Table D.4: Sensitivity of the set-size to the size of $\mathcal{D}_{\text{cells}}$. We select a subset $\mathcal{D}^{p}_{\text{cells}} \subseteq \mathcal{D}_{\text{cells}}$, where $|\mathcal{D}^{p}_{\text{cells}}| = p \cdot |\mathcal{D}_{\text{cells}}|$. The set sizes obtained for different values of $p$ are reported for CIFAR100 dataset and 7 heads.

| Method | Score | $\alpha$ | Proportion | | | | | | | | | |
|---|---|---|---|---|---|---|---|---|---|---|---|---|
| | | | 100% | 95% | 90% | 85% | 80% | 70% | 60% | 50% | 25% | 10% |
| Multi-score | RAPS | 0.1 | 15.32 | 15.66 | 15.68 | 15.69 | 15.92 | 16.26 | 16.68 | 17.32 | 19.04 | 22.13 |
| | | 0.2 | 2.33 | 2.33 | 2.34 | 2.35 | 2.37 | 2.49 | 2.54 | 2.64 | 3.09 | 4.16 |
| | SAPS | 0.1 | 15.46 | 15.46 | 15.76 | 15.72 | 15.88 | 16.04 | 16.32 | 16.91 | 19.01 | 22.17 |
| | | 0.2 | 2.36 | 2.36 | 2.37 | 2.41 | 2.42 | 2.54 | 2.6 | 2.64 | 3.09 | 3.61 |
| Optimized | RAPS | 0.1 | 22.47 | 22.27 | 22.04 | 22.09 | 21.85 | 21.71 | 22.47 | 22.52 | 23.4 | 24.2 |
| | | 0.2 | 6.53 | 6.31 | 6.21 | 6.3 | 6.73 | 6.49 | 6.14 | 6.55 | 6.54 | 7.21 |
| | SAPS | 0.1 | 21.8 | 21.83 | 21.71 | 21.68 | 21.51 | 21.63 | 21.55 | 21.66 | 23.24 | 25.11 |
| | | 0.2 | 6.8 | 6.45 | 6.4 | 6.45 | 6.31 | .6.14 | 5.93 | 5.93 | 7.4 | 8.611 |

used, i.e., we select a subset $\mathcal{D}^{p}_{\text{cells}} \subseteq \mathcal{D}_{\text{cells}}$, where $|\mathcal{D}^{p}_{\text{cells}}| = p \cdot |\mathcal{D}_{\text{cells}}|$. The set sizes obtained for different values of $p$ are presented in Table D.4. As expected, the set size increases as $p$ decreases. However, the overall behavior remains stable, with only $9.4 - 13.3\%$ increase in set size for $50\%$ reduction in the number of samples in $\mathcal{D}_{\text{cells}}$. In addition, we compare to the Optimized baseline, where the set $\mathcal{D}^{p}_{\text{cells}}$ is used for optimizing the weights for combining the scores. We see for all $p$ values Multi-score is advantageous over Optimized.

**Influence of the nonconformity score's parameters .** We examined the influence of the different parameters of the non-conformal RAPS and SAPS scores on the methods performances . The results for the RAPS score are detailed in Table D.5, while those for the SAPS score are presented in Table D.6. Overall, our findings indicate that the Multi-Score method remains consistently advantageous, regardless of the parameter settings.

**Influence of the underlying model.** To demonstrate that our method improves the efficiency of conformal prediction (CP) regardless of the underlying model, we conducted an experiment using a more powerful ViT model Dosovitskiy et al. (2020) Results are presented on table D.7, showing similar trends to those observed in our main results.

## D.4 RESULTS ON OTHER DATASETS

**Results on text data classification.** To show that our method can be applied across different types of data, we conducted an experiment with a text classification task. We use the 20 Newsgroups dataset, which comprises newsgroup posts on 20 topics. We use a BERT-base model (Devlin et al., 2019), and attach additional classification heads in a similar way to the other models. The results on Table D.8 show that Multi-Score outperforms all the baselines. Here, the norm baseline is the closest in the performance to the proposed method.

**Results for ImageNet.** Table D.9 presents the results obtained for ImageNet. Here the classification heads consist of a single linear layer, and are all initialized by the weights of the pretrained model. Here too we can see the benefit of the proposed method over the baselines.

## D.5 ADDITIONAL BASELINES

**L1 Norm baseline.** We examined an additional baseline, where we use the L1 norm instead of the L2 norm. Table D.10 compares the two baselines for CIFAR-100 dataset and RAPS score. We observe that both baselines obtain similar performance.

**Set-size comparison to vanilla baseline.** We examined a vanilla baseline, where the CP procedure is performed over the original classification head, without the addition of multiple classification heads. We observe that the results are similar to the Best head baseline defined above. Table D.11 summarizes the results

Table D.5: Set-size comparison between the baselines for different values of $\kappa$ and $\lambda$, used in RAPS score, on CIFAR100, $\alpha = 0.1$ and 7 heads.

| Method | $\kappa$ | $\lambda = 0.001$ | $\lambda = 0.01$ | $\lambda = 0.1$ | $\lambda = 1.0$ |
|---|---|---|---|---|---|
| Multi-score | 1 | 12.79 | 14.83 | 15.10 | 15.45 |
| | 2 | 12.84 | 14.26 | 15.11 | 15.53 |
| | 3 | 12.88 | 14.29 | 15.30 | 15.54 |
| | 4 | 12.81 | 14.45 | 15.36 | 15.53 |
| | 5 | 12.80 | 14.44 | 15.51 | 15.70 |
| Best head | 1 | 32.04 | 32.11 | 32.11 | 32.11 |
| | 2 | 24.41 | 24.00 | 24.00 | 25.00 |
| | 3 | 23.01 | 23.02 | 23.04 | 22.79 |
| | 4 | 20.77 | 21.12 | 21.63 | 22.70 |
| | 5 | 20.32 | 21.21 | 23.60 | 22.60 |
| Uniform | 1 | 24.38 | 24.76 | 24.91 | 24.96 |
| | 2 | 24.41 | 23.59 | 24.87 | 25.05 |
| | 3 | 23.10 | 23.75 | 25.44 | 26.14 |
| | 4 | 21.20 | 21.84 | 23.26 | 23.33 |
| | 5 | 20.40 | 21.41 | 22.63 | 22.75 |
| Optimized | 1 | 24.45 | 24.89 | 25.26 | 25.84 |
| | 2 | 24.52 | 24.00 | 24.76 | 25.10 |
| | 3 | 23.09 | 23.32 | 24.64 | 25.07 |
| | 4 | 20.82 | 21.20 | 22.30 | 22.70 |
| | 5 | 20.34 | 21.40 | 22.80 | 22.60 |
| Norm | 1 | 18.01 | 19.62 | 19.80 | 22.59 |
| | 2 | 18.01 | 22.99 | 24.53 | 24.66 |
| | 3 | 22.82 | 23.05 | 27.74 | 28.50 |
| | 4 | 20.80 | 21.21 | 21.75 | 23.60 |
| | 5 | 20.34 | 21.21 | 23.86 | 25.04 |

Table D.6: Set-size comparison between the baselines for different values of $\xi$, used in SAPS score, on CIFAR100, $\alpha = 0.1$ and 7 heads.

| Method | $\xi = 0.01$ | $\xi = 0.05$ | $\xi = 0.1$ | $\xi = 0.5$ | $\xi = 1.0$ |
|---|---|---|---|---|---|
| Multi-score | 14.70 | 16.17 | 16.28 | 16.89 | 17.06 |
| Best head | 34.49 | 35.11 | 35.21 | 35.43 | 35.48 |
| Uniform | 19.15 | 21.73 | 21.91 | 22.55 | 22.57 |
| Optimized | 18.51 | 21.00 | 21.60 | 21.90 | 22.07 |
| Norm | 18.61 | 21.31 | 23.12 | 24.56 | 24.59 |

Table D.7: ResNet vs. ViT model set size comparison on CIFAR100 with $\alpha = 0.1$ and 7 heads.

| Model | Acc. | Score | Method | | | | |
|---|---|---|---|---|---|---|---|
| | | | Multi-score | Best head | Uniform | Optimized | Norm |
| ViT | 0.8 | RAPS | 4.34 | 22.11 | 16.21 | 15.93 | 5.51 |
| | | SAPS | 2.33 | 12.77 | 2.43 | 2.45 | 2.60 |
| ResNet50 | 0.69 | RAPS | 15.32 | 35.29 | 22.37 | 22.47 | 22.49 |
| | | SAPS | 15.46 | 35.41 | 22.05 | 21.80 | 22.79 |

Table D.8: Set-size comparison across different methods on the 20 Newsgroups dataset, with 7 heads, using the BERT-base model fine-tuned for news topic classification.

| $\alpha$ | Score | Method | | | | |
|---|---|---|---|---|---|---|
| | | Multi-score | Best head | Uniform | Optimized | Norm |
| 0.08 | RAPS | 1.46 | 9.61 | 9.58 | 9.49 | 1.5 |
| | SAPS | 1.47 | 10.66 | 2.09 | 2.18 | 2.35 |
| 0.05 | RAPS | 2.29 | 10.1 | 10.01 | 10.07 | 2.32 |
| | SAPS | 1.97 | 10.9 | 2.4 | 2.46 | 2.62 |
| 0.02 | RAPS | 4.28 | 11.73 | 11.73 | 11.47 | 4.36 |
| | SAPS | 3.28 | 11.57 | 3.45 | 3.34 | 3.65 |

Table D.9: Set-size comparison across different methods on the ImageNet dataset, with 7 heads, demonstrating the performance with a large-number-of classes.

| $\alpha$ | Score | Method | | | | |
|---|---|---|---|---|---|---|
| | | Multi-score | Best head | Uniform | Optimized | Norm |
| 0.1 | RAPS | 4.1 | 4.36 | 13.47 | 14.63 | 13.47 |
| | SAPS | 4.94 | 5.36 | 13.85 | 14.99 | 13.85 |
| 0.2 | RAPS | 1.82 | 2.4 | 2.1 | 2.1 | 2.1 |
| | SAPS | 1.57 | 1.68 | 2.04 | 2.06 | 2.04 |

Table D.10: Set-Size comparison between L1 and L2 Norm methods on CIFAR100 and 7 heads.

| $\alpha$ | Score | Method | | |
|---|---|---|---|---|
| | | L1 | L2 | Multi-score |
| 0.1 | RAPS | 22.98 | 22.49 | 15.32 |
| | SAPS | 22.25 | 22.79 | 15.46 |
| 0.2 | RAPS | 6.75 | 7.51 | 2.33 |
| | SAPS | 7.62 | 7.85 | 2.36 |

Table D.11: Set-size comparison between the vanilla baseline and the multi-score methods on CIFAR100, with RAPS score and 7 heads.

| $\alpha$ | Set Size | | |
|---|---|---|---|
| | Multi-score | Vanilla RAPS | Best head |
| 0.1 | 15.32 | 39.27 | 35.29 |
| 0.2 | 2.33 | 8.09 | 9.37 |

## D.6 Variants of Multi-Score Conformal Prediction

**Jackknife+ Multi-score conformal prediction.** We compare our split version in Algorithm 1 to the jackknife+ version in Algorithm B.1. We obtained that the required 1-$\alpha$ coverage is achieved in both settings, and the set sizes are summarized in Table D.12. As expected jackknife+ obtains smaller set sizes. However, the improvement appears to be insignificant in this case and may not justify the additional computational cost.

Table D.12: Comparison between Jackknife+ and Multi-score methods on CIFAR100 and 7 heads.

| $\alpha$ | Score | Method | |
|---|---|---|---|
| | | Multi-score Jackknife+ | Multi-score |
| 0.1 | RAPS | 14.85 | 15.32 |
| | SAPS | 14.61 | 15.46 |
| 0.2 | RAPS | 2.26 | 2.33 |
| | SAPS | 2.22 | 2.36 |

Table D.13: Comparison of soft centers approach with different number of neighbors $b$ on CIFAR100 and 7 heads.

| $\alpha$ | Score | Method | | | | |
|---|---|---|---|---|---|---|
| | | $b = 200$ | $b = 100$ | $b = 50$ | $b = 10$ | Multi-score ($b = 1$) |
| 0.1 | RAPS | 23.68 | 22.73 | 19.32 | 16.48 | 15.32 |
| | SAPS | 23.53 | 22.44 | 19.12 | 17.14 | 15.46 |
| 0.2 | RAPS | 3.01 | 2.89 | 2.75 | 2.26 | 2.33 |
| | SAPS | 3.04 | 2.87 | 2.81 | 2.29 | 2.36 |

**Soft Multi-score conformal prediction.** We evaluate the soft version of our proposed approach. The set sizes obtained for different number of neighbors are summarized in Table D.13. We observe that in almost all cases $b = 1$ (Algorithm 1) is preferable.

