# OpenReview forum: "Multi-Dimensional Conformal Prediction"
_ICLR.cc/2025/Conference — ICLR 2025 Poster_

### Official Review · Reviewer_9Lus · 2024-10-27

**Soundness:** 3
**Presentation:** 3
**Contribution:** 2
**Rating:** 5
**Confidence:** 3

**Summary:**

The paper introduces a multi-dimensional conformal prediction method for classification problems. A nonconformity measure is defined depending on the multiple scores, which leads to finer separation between correct and wrong labels. The proposed procedure mainly includes three steps: first, partitioning the score space into cells using $D_{cells}$; second, scoring and ranking the cells according to the ratio of false to true labels; and third, calibration-achieved by progressively adding top-ranked cells until the desired coverage level is reached.

**Strengths:**

The paper is well-organized. The problematic and approach are well introduced, and the concept concisely presented. The proposed method successfully extends conformal prediction into a multi-dimensional setting with enhanced separation between correct and incorrect labels, and it appears sufficiently general.

**Weaknesses:**

1. The experiments mainly focus on image classification datasets (e.g., CIFAR100, Tiny ImageNet, PathMNIST), limiting the demonstration of the other data types, such as text or tabular data.

2. While the paper is claimed to be cost-efficient, it still involves training and managing multiple classification heads, which could be computationally expensive for large-scale datasets or resource-constrained environments. Could you provide more insights into the computational cost? Specifically, how does it compare to other baseline methods?

**Questions:**

How sensitive is the method to hyperparameter choices, such as the number of classification heads or diversity regularization? And how do different configurations affect set size?

---

> ### Author Response · Authors · 2024-11-24
> **Response to Reviewer 9Lus**
>
> Thank you for your helpful and constructive review.
>
> > On demonstration over other data types
>
> Thank you for your suggestion. Following your comment, we added an experiment with a text classification task in Appendix D.4.  We use the 20 Newsgroups dataset, which comprises newsgroup posts on $20$ topics. We use a BERT-base model, and attach additional classification heads in a similar way to the other models. The results are reported on Table D.7:
>
> | **α** | **Score** | Multi Score | Best Head | Uniform | Optimized | Norm |
> |-------|-----------|-------------|-----------|----------|-----------|------|
> | 0.08  | RAPS      | 1.46        | 9.61      | 9.58     | 9.49      | 1.5  |
> | 0.08  | SAPS      | 1.47        | 10.66     | 2.09     | 2.18      | 2.35 |
> | 0.05  | RAPS      | 2.29        | 10.1      | 10.01    | 10.07     | 2.32 |
> | 0.05  | SAPS      | 1.97        | 10.9      | 2.4      | 2.46      | 2.62 |
> | 0.02  | RAPS      | 4.28        | 11.73     | 11.73    | 11.47     | 4.36 |
> | 0.02  | SAPS      | 3.28        | 11.57     | 3.45     | 3.34      | 3.65 |
>
> Similarly to the other dataset, we can see the benefit of the proposed method over the baselines.
>
> > On providing more insights into the computational cost? Specifically, how does it compare to other baseline methods?
>
> Thank you for pointing this out. Our proposed self-ensemble model requires attaching multiple classification heads to the model. Let $P$ denote the number of parameters per head and $H$ denote the number of heads, than the model size is increased by $P\cdot(H-1)$ parameters. in practice, we used $2$ fully connected layers for  each classification head. Our results indicate that there is already an improvement with only $2$ classification heads, and that it is usually unnecessary to add more than $4$ heads. Thus, the size of the model is not largely affected by this modification.  In terms of inference time, this is exactly as using a standard model with a single head, as all heads can be computed in parallel.
>
> It should be emphasized that the main idea in the paper is how we can use multiple nonconformity scores to improve conformal prediction efficiency, where we propose to perform the calibration in the high-dimensional space instead of averaging the scores and performing standard one-dimensional conformal prediction.  This idea can be applied in the case we have multiple different scores, which can be obtained in different ways: standard ensemble of multiple different models, test-time augmentation where scores are computed for the output of different augmented versions of the input image, multiple types of scores computed from a single output (e.g. Thr, APS, RAPS, SAPS) and our proposed self-ensemble that extracts scores from multiple classification heads. Note that test time augmentation does not require any modifications to the base model, and a single head with different type of scores is the least expensive option.  Our main results are shown for the self-ensemble approach, while we also provide results for the other cases.
>
> > On the sensitivity to hyperparameter choices, such as the number of classification heads or diversity regularization.
>
> **Sensitivity to the number of heads** - We can see on Figs. 3, D.1, D.2 and D.3 that as the number of heads increases we obtain smaller set sizes. However, an improvement is already observed for two heads, and most of the time $4-5$ heads are enough to get the optimal results.
>
> **Sensitivity to the diversity regularization** - Following your comment, we added an experiment analyzing the influence of the parameter controlling the weight of the diversity regularization in Appendix D.3.:
>
> "We conducted an ablation study to examine the affect of adding diversity regularization to the loss used for training the classification heads, defined in Eq. (12). We use the same fine-tuned model in the first training stage and change only the second stage to optimize the regularized loss in Eq. (12) with different values of $\lambda$, controlling the strength of the diversity regularization. Figure D.10.
> shows the similarity between heads with and without the diversity regularization. As expected, the similarity between heads is smaller for the model trained with the regularized loss. Figure D.11. compares the set sizes obtained for different values of $\lambda$, demonstrating that adding the regularization results in smaller sets. The optimal value of $\lambda$ is around $0.8$, after which an increase in set size is observed, apparently due to the fact that increasing head diversity comes at the cost of decreasing the accuracy of each head."

---

> ### Author Response · Authors · 2024-12-01
>
> Dear reviewer 9Lus,
>
> Thank you very much again for your helpful and constructive feedback. Your insights have been invaluable in refining our work, and we are grateful for the time and effort you have dedicated to reviewing our manuscript.
>
> As the review process deadline approaches, we kindly request that you revisit our submission in light of the revisions and corrections we have made in response to your valuable feedback. We have carefully addressed the points you raised and believe these changes have strengthened the paper significantly.
>
> If you find the revisions satisfactory, we would greatly appreciate it if you could consider raising your score to reflect the improvements.
> We would be more than happy to provide further clarifications if you have any additional concerns.
>
> Best,
> Authors

---

### Official Review · Reviewer_UGXT · 2024-10-28

**Soundness:** 3
**Presentation:** 3
**Contribution:** 3
**Rating:** 8
**Confidence:** 3

**Summary:**

This paper tends to combine several different conformity score functions aiming to reduce the prediction set size. In short their approach is to combine different score functions per $(x, y)$ forming a vector of scores. Dividing the calibration set into two separate sets of re-cal and tuning set, they use the tuning set as the basis (centroids) in the mutli-dimensional scores space. They assign a rating to each of these centriods based on the proportion of false labels inside that area (this means 1 + set size if there is only one point at the place). During calibration they deactivate regions in decreasing order to the number of false predictions, as long as the $1 - \alpha$ coverage is satisfied. Each calibration point is assigned to the closest centriod. Similarly during test. they check the closest centiod and add the label to the prediction set if the centriod is active.

The propose different methods to build multi-dimensional score including using different classification heads, to simply using different score functions.

**Strengths:**

The method is simple yet effective. The paper is also easy to read and “to the point”. I do not see any major flaw or discouraging point in it. I also see that the authors tried to include a comprehensive evaluation by comparing with best head, uniform average and optimized weights.

**Weaknesses:**

1. Why the authors did not report worst slice coverage which is metric of adaptivity?
2. Why the authors used an unnecessarily large calibration set? I don’t think the evaluation setup is realistic with 16500 calibration points to predict 11000 test points. What is the gain of this large calibration set? Is there any statistical efficiency results that are missing?
3. Related to the previous problem, a fair comparison is to report paper’s best method in comparison with the normal best method where the tuning set is also added to the calibration set. The results are not expected to be significantly different but still this is important to be reported.
4. Why the authors did not report their result with APS and threshold prediction sets? Although there are adaptivity and efficiency constraints, still those result are important to be reported. RAPS and SAPS are ignoring informations in the logit space either by penalty, or just reporting the rank, and intuitively in a perfect classifier this should cause inefficiencies.
5. Is there any preference to have 0-1 assignment to the zones?, the author could also try soft probability assignment to each region and optimize for that. It is not a must, but either trying soft centroid assignment, or showing that the optimal is happening with the current 0-1 assignment would make the work stronger.

I would increase my score if a convincing answer to these two questions is provided.

1. Why the calibration set is very large in all experiments. Are there any limitations? What would the result be with smaller calibration set (e.g. 2000 points)
2. Why threshold prediction sets (softmax score) and adaptive prediction sets are not evaluated. The authors could have similar results with these score functions as well.
3. An ablation study on the number of centroids is important. How the results would change with very limited number of centriod (e.g. 200)?

**Questions:**

Please see Weaknesses.

---

> ### Author Response · Authors · 2024-11-24
> **Response to Reviewer UGXT - Part I**
>
> Thank you for your helpful and constructive review.
>
> > On reporting worst slice coverage as a metric of adaptivity
>
> We report the worst slice coverage (the size-stratified coverage violation (SSCV) metric defined in Eq. (18)) in Table D.1:
>
> | **α** | **Head** | **Best head** | **Multi-score** | **Norm** | **Optimized** | **Uniform** |
> |-------|----------|---------------|-----------------|-----------|---------------|-------------|
> | 0.1   | 1        | 0.174         | 0.034          | 0.174     | 0.159         | 0.174       |
> | 0.1   | 2        | 0.167         | 0.108          | 0.114     | 0.194         | 0.141       |
> | 0.1   | 3        | 0.240         | 0.104          | 0.066     | 0.162         | 0.277       |
> | 0.1   | 4        | 0.298         | 0.204          | 0.178     | 0.167         | 0.264       |
> | 0.1   | 5        | 0.106         | 0.155          | 0.167     | 0.200         | 0.252       |
> | 0.1   | 6        | 0.131         | 0.153          | 0.172     | 0.308         | 0.125       |
> | 0.1   | 7        | 0.056         | 0.159          | 0.062     | 0.191         | 0.175       |
> | 0.2   | 1        | 0.190         | 0.037          | 0.190     | 0.157         | 0.190       |
> | 0.2   | 2        | 0.362         | 0.249          | 0.292     | 0.221         | 0.330       |
> | 0.2   | 3        | 0.342         | 0.200          | 0.132     | 0.187         | 0.098       |
> | 0.2   | 4        | 0.379         | 0.206          | 0.167     | 0.166         | 0.279       |
> | 0.2   | 5        | 0.361         | 0.192          | 0.159     | 0.209         | 0.331       |
> | 0.2   | 6        | 0.220         | 0.180          | 0.071     | 0.274         | 0.276       |
> | 0.2   | 7        | 0.186         | 0.203          | 0.200     | 0.171         | 0.245       |
>
> We can see that all methods achieve similar SSCV scores, with no method showing a clear advantage over the others.
>
> Note that our cell score, defined by Eq. (8), puts an emphasis on the efficiency in terms of set size, however we could prioritize cells according to different metrics, for example using a score that takes into account some grouping of the input $x$ to encourage adaptivity. This is indeed an interesting direction for future work.
>
> > On using unnecessarily large calibration set
>
> Thank you. We added an experiment comparing the performance with varying sizes of the calibration set in Appendix D.3. We vary the proportion $p$ of samples from the defined calibration split that are actually used. Then, the this $p$-th subset is split into half for cell computation, and re-calibration.  Figure D.12 presents the set sizes for different values of $p$. It can be seen that the proposed method is almost always preferable, with its advantage becoming more pronounced as the size of the calibration data increases.
>
> In addition, following the comment of reviewer 4GpN, we added a leave-one-out version of the proposed method that is more data efficient, as it does not require splitting the calibration data into two subsets. For more details, see Section 4.2. and Algorithm B.1.
>
> > On comparison with the normal best method where the tuning set is also added to the calibration set
>
> Thank you for pointing this out. We agree that a fair comparison should include the normal method where the tuning set is also added to the calibration set, and this is indeed the baseline that we use. We added a clarification in the description of the baseline methods.
>
> > On reporting result with APS and threshold prediction sets
>
> Thank you for pointing this out. Following your comment, we added experiments, evaluating our method over APS and threshold prediction sets in Appendix D.1:
>
> “Our proposed method can be applied over any type of score. However, we have seen that it is not fully optimal for Thr and APS. The reason is that for these scores the values are more concentrated on specific levels, while for SAPS and RAPS the values are more spread. In order to improve this behavior we use temperature scaling, i.e. we divide the logits by $T$ before applying Softmax(). As $T$ increases the entropy of the probabilities increases and they become more spread, as illustrated in Fig. D.5. Figure D.6. presents the set sizes obtained for all methods with respect to different temperatures. We see that the efficiency the proposed method greatly improves as $T$ increases for both Thr and APS, while the performance of the baseline methods is less effected by $T$. For RAPS and SAPS the proposed method outperforms the baselines regardless of the temperature due to the inherent spread of these scores.”

---

> ### Author Response · Authors · 2024-11-24
> **Response to Reviewer UGXT - Part II**
>
> > On trying soft probability assignment to each region and optimize for that
>
> Thank you for your suggestion. We added a new version of our method based on your proposal in Section 4.2:
>
> "We present a more general variant of our approach, where instead of considering a hard assignment of each point into the closest center, we consider a softer assignment to $b$ nearest neighbors. This variant is summarized in Algorithm B.2. Here we detect the $b$ nearest neighbors, and include the point in the prediction set if at least half of its neighbors are in the chosen region."
>
> We report the results in Table D.12:
>
> | **α** | **Score** | **Methods** | | | | |
> |-------|-----------|-------------|-------------|-------------|-------------|-----------------|
> | | | **b=200** | **b=100** | **b=50** | **b=10** | **Multi Score (b=1)** |
> | 0.1 | RAPS | 23.68 | 22.73 | 19.32 | 16.48 | 15.32 |
> | | SAPS | 23.53 | 22.44 | 19.12 | 17.14 | 15.46 |
> | 0.2 | RAPS | 3.01 | 2.89 | 2.75 | 2.26 | 2.33 |
> | | SAPS | 3.04 | 2.87 | 2.81 | 2.29 | 2.36 |
>
> We observe that in almost all cases $b=1$ (Algorithm 1) is preferable.
>
> > On an ablation study over the number of centroids.
>
> We extended the ablation study over the number of centroids to show the influence of a very limited number of centroids (Appendix D. 3., Table D.3.):
>
> | | | **α** | 100% | 95% | 90% | 85% | 80% | 70% | 60% | 50% | 25% | 10% |
> |---|---|---|---|---|---|---|---|---|---|---|---|---|
> | **Multi Score** | RAPS | 0.1 | 15.32 | 15.66 | 15.68 | 15.69 | 15.92 | 16.26 | 16.68 | 17.32 | 19.04 | 22.13 |
> | | | 0.2 | 2.33 | 2.33 | 2.34 | 2.35 | 2.37 | 2.49 | 2.54 | 2.64 | 3.09 | 4.16 |
> | | SAPS | 0.1 | 15.46 | 15.46 | 15.76 | 15.72 | 15.88 | 16.04 | 16.32 | 16.91 | 19.01 | 22.17 |
> | | | 0.2 | 2.36 | 2.36 | 2.37 | 2.41 | 2.42 | 2.54 | 2.6 | 2.64 | 3.09 | 3.61 |
> | **Optimized** | RAPS | 0.1 | 22.47 | 22.27 | 22.04 | 22.09 | 21.85 | 21.71 | 22.47 | 22.52 | 23.4 | 24.2 |
> | | | 0.2 | 6.53 | 6.31 | 6.21 | 6.3 | 6.73 | 6.49 | 6.14 | 6.55 | 6.54 | 7.21 |
> | | SAPS | 0.1 | 21.8 | 21.83 | 21.71 | 21.68 | 21.51 | 21.63 | 21.55 | 21.66 | 23.24 | 25.11 |
> | | | 0.2 | 6.8 | 6.45 | 6.4 | 6.45 | 6.31 | 6.14 | 5.93 | 5.93 | 7.4 | 8.611 |
>
> As expected, the set size increases as $p$ decreases. However, the overall behavior remains stable, with only $9.4\\%-13.3\\%$ increase in set size for $50\\%$ reduction in the number of cells. In addition, we compare to the Optimized baseline, where the same set is used for optimizing the weights for combining the scores. We see for all $p$ values Multi-score method is advantageous over the Optimized baseline.

---

> > ### Comment · Reviewer_UGXT · 2024-11-25
> >
> > Thank you for addressing my concerns and for implementing new experiments. I am happy to increase my score, which suggests the paper's acceptance.
> >
> > Also additional questions. I do not expect major changes in the paper, but I want to recall if these questions are addressed or not:
> > 1. Does using several scores at once introduce any statistical flaws like multiple hypothesis testing or seeing a statistical variable twice?
> > 2. Do the authors suggest any intuitive understanding on why designing this grid and selecting cells help? Is there any lost information that is retrieved with this procedure? Does the system of multiple scores increase the uncertainty quantification quality?

---

> > > ### Author Response · Authors · 2024-11-25
> > > **Thanks for your response**
> > >
> > > Thank you very much for reviewing our response and for your feedback.
> > >
> > > > 1. Does using several scores at once introduce any statistical flaws like multiple hypothesis testing or seeing statistical variable twice?
> > >
> > > Thank you for pointing this out. This is indeed true that using several scores at once might cause statistical flaws. For example, constructing a prediction set per score and taking the intersection of sets is akin to a multiple hypothesis testing problem. However, our approach avoids these issues by combining the different scores into a multi-dimensional representation and performing a single unified calibration, rather than calibrating each axis separately.
> > >
> > > In addition, we carefully split the calibration data into two parts where one is used for cell portioning and ranking and the other is reserved for selecting cells that obtain the desired coverage. This way there is no problem of seeing a statistical variable twice, as we rigorously show in the proof provided in Appendix A.2. Moreover, the Jackknife+ variant, which was added to the revised manuscript (see Algorithm B.1.), maintains validity without data splitting using a leave-one-out strategy.
> > >
> > > > 2. Do the authors suggest any intuitive understanding on why designing this grid and selecting cells help? Is there any lost information that is retrieved with this procedure? Does the system of multiple scores increase the uncertainty quantification quality?
> > >
> > > The cell partitioning is similar in principle to one dimensional conformal prediction, where the calibration scores define segments of varying lengths, and the final selected interval is the union of all segments to the left of the computed quantile. Thus, our cell partitioning method can be seen as a generalization of the standard partitioning process to higher dimensions. However, unlike the one-dimensional case, determining which cells (instead of segments) should be included in the prediction set is not straightforward. Although it is clear that scores with low value in all dimensions correspond to the conforming labels, there are still infinitely many selection possibilities that achieve the desired coverage. Therefore, we define the cell score in Eq. (9), prioritizing cells with low number of false labels to encourage small and tight prediction sets. Selecting the subset of cells that covers $(1-\alpha)$ proportion of the second calibration split results in a selection region that is both valid and expected to yield prediction sets with the least number of false labels. In general, compared to the standard one-dimensional case, higher-dimensional spaces can achieve better separation between scores of correct and incorrect labels, potentially leading to more efficient prediction sets.
> > >
> > > Moreover, there are indeed cases where using only a single score may loose information and the system of multiple scores can retrieve such information and increase the uncertainty quantification quality. In our revised manuscript, we included a synthetic example to illustrate this scenario (see Section 4.1). In this example, two classifiers approximate the conditional distribution $P(y|x)$ accurately for specific parts of the $x$-domain, while being noisy in other regions. Notably, these regions differ between the two classifiers. Thus, each classifier provides perfect uncertainty information for only a subset of the input $x$. Combining the two-dimensional scores from both classifiers fills up the missing information across the entire $x$ domain (see Figure 2 for illustrations). This example also provides the motivation for our self-ensemble approach, where multiple classification heads are trained with a diversity-regularized loss. This loss encourages classifiers to produce complementary predictions by specializing in specific input domains. This specialization suggests that, when considered together in a multi-dimensional space, the classifiers' scores can offer improved uncertainty quantification. Following your comment, we added this clarification to the revised manuscript.

---

> ### Comment · Reviewer_UGXT · 2024-11-29
> **Possible mistake!**
>
> I think the authors misunderstood my question (Q1). Although I do not decrease my score, this is an important concern. Upon justification and a report on this metric, I will revisit the paper to see if I can increase the score again.
>
> Here as a measure of conditional coverage, I referred to worst-slice coverage not set-size stratified coverage. For the worst slice coverage, I refer to [1] (section S.1.2 in the appendix). To quantify that, we create $k$ random vectors (tensors) with the dimensionality as the data. Then we sample $a, b$ randomly but among the sorted values of dot product results. This basically captures that on a random projection, within a bound what is the worst coverage. This metric captures if a local region in the distribution is miscovered (regardless of the set size).
>
> **Why is this metric important?** The size-stratified coverage (worst coverage among sets of the same size) does not capture the information stored in conformal scores and how it quantifies uncertainty. Assume a dataset in which all points are aleatorically uncertain. Here small set sizes are misleading since if the score captures this uncertainty for every point all sets are large.
>
> Another justification is to assume that our dataset has a minority subgroup for which the conformal prediction is miscovering.  In that group the set size could be anything, and maybe reporting the coverage over sets of size 1, 2, ... has the same averaging effect that the marginal coverage has. Because the group is treated unfairly w.r.t. coverage not the average set size. While the worst slice coverage as defined in [1] is more likely to capture this notion.
>
> The SAPS score function (Huang et al., 2024 in your references) also reports the set-size stratified coverage as a notion of conditional coverage quantification which is completely wrong as it does not incorporate any information regarding the location of $x$ in any data manifold.
>
> Even if your method on the worst-slice coverage is imbalanced I would not decrease my score, but I expect it to be mentioned in the limitations.
>
> [1] Classification with Valid and Adaptive Coverage, Yaniv Romano, Matteo Sesia, Emmanuel J. Candès, 2020.

---

> > ### Author Response · Authors · 2024-11-30
> >
> > Thank you for your valuable feedback and for highlighting the distinction between set-size stratified coverage and worst-slice coverage (WSC). We calculated the WSC metric as defined in [1], section S.1.2, and the results, averaged over 10 trials, are presented in the table below:
> >
> >
> > | **α**   | **Dataset**      | **Score** | **Multi Score** | **Best Head** | **Uniform** | **Optimized** | **Norm** |
> > |---------|------------------|-----------|-----------------|---------------|-------------|---------------|----------|
> > | 0.1     | CIFAR100         | RAPS      | 0.901           | 0.905         | 0.903       | 0.905         | 0.901    |
> > |         |                  | SAPS      | 0.903           | 0.908         | 0.906       | 0.895         | 0.894    |
> > |         | TinyImageNet     | RAPS      | 0.905           | 0.903         | 0.896       | 0.896         | 0.892    |
> > |         |                  | SAPS      | 0.900           | 0.902         | 0.900       | 0.901         | 0.909    |
> > | 0.2     | CIFAR100         | RAPS      | 0.801           | 0.801         | 0.797       | 0.799         | 0.803    |
> > |         |                  | SAPS      | 0.812           | 0.819         | 0.800       | 0.813         | 0.817    |
> > |         | TinyImageNet     | RAPS      | 0.796           | 0.791         | 0.801       | 0.800         | 0.802    |
> > |         |                  | SAPS      | 0.808           | 0.802         | 0.808       | 0.806         | 0.807    |
> >
> >
> >
> >
> > These results show that the multi-score method performs comparably to the baselines in terms of WSC, offering additional insights regarding the performance of our approach in terms of the conditional coverage. We will ensure that this metric is included in paper during the next editing phase.
> >
> > Thank you again for raising this important point. We believe this addition strengthens the paper and aligns with your concerns.

---

> > > ### Comment · Reviewer_UGXT · 2024-12-01
> > >
> > > Thanks for evaluating the worst slice coverage. I would be happy if the final version of the paper includes these results + an explanation of the observation.
> > >
> > > In light of all the changes I believe the work is not in the marginal zone. I understand that increasing the score from 6 to 8 makes a contrast with other reviewers especially due to the rejected score. I read the comments and I understood that the main concern for rejection is the lack of explanation on why this multi-view helps CP in efficiency.
> > >
> > > An explanation for this requires either showing that the new perspective carries more information about the uncertainty quantification or showing that the rankings are more aligned with ground truth aleatoric uncertainty. The authors can observe the latter via synthetic experiments. However, throughout the literature,  I still do not see any direct connection between CP and UQ in general. Therefore such tasks on its own is a separate contribution outside of this scope.
> > >
> > > With all of the abovementioned and all the updates made by the authors, I believe that the work is now in a high quality for the conference. However, for future, and in general, I do not expect other works to compare with this approach as a baseline. To the best of my understanding, this work is more about incorporating more information to have a  less denoised ranking for CP.

---

### Official Review · Reviewer_52ya · 2024-11-03

**Soundness:** 1
**Presentation:** 4
**Contribution:** 2
**Rating:** 3
**Confidence:** 4

**Summary:**

In this paper, the authors aim to improve the efficiency of conformal prediction sets with multiple non-conformity scores. In particular, they propose a multidimensional framework of conformal predictions, which uses a single score in one of the dimensions. They provide a theory to show the desired coverage can be guaranteed through the multidimensional selection. To obtain multiple diverse nonconformity scores, they employ a multi-head model and train it with a diversity regularization. The proposed method is validated on a CIFAR100, Tiny ImageNet, and PathMNIST to show the effectiveness.

**Strengths:**

1. The method is novel. To the best of my knowledge, this is the first work to utilize ensemble predictions as multidimensional score.
2. The empirical improvement is significant. The proposed method can produce smaller prediction sets compared to other baselines.

**Weaknesses:**

Major:
1. Why the multi-score conformal calibration can work better is not explained. The authors only provide a theorem to ensure the valid coverage of the proposed method, but do not elaborate about how it leads to efficient prediction sets. It could be better if they can provide convincing explanation in this aspect.

2, The multi-score setting seems different from previous works. Previous multi-score works aim to utilize multiple score functions to obtain better performance, but this work proposes the method to generate multiple scores with ensemble models. In my view, this problem should be CP for ensemble models instead of the multi-score setting. So it might be better for authors to rewrite the motivation (subsection 3.2) for avoiding misunderstanding. (Or the authors can show the method can also improve the setting of multiple score functions.)

3. The empirical comparison is not extensive. The authors should include ImageNet in the experiments, or it is hard to evaluate the effectiveness of the proposed method in large number of classes.

Minor:
1. presentation: it is hard to read the information from Figures 2 and 3. Maybe the authors can simply provide tables to show the results.
2. It might be better to put some results in Appendix B in the main part, as there is almost one page left as blank.

**Questions:**

In figure 2, are all the compared methods employed on the multi-head model trained by the loss in Equation (11)?

---

> ### Author Response · Authors · 2024-11-24
> **Response to Reviewer 52ya - Part I**
>
> Thank you for your helpful and constructive review.
>
> >  On explaining why the multi-score conformal calibration can work better
>
> Thank you. Following your and reviewer 4GpN comments,  we added a motivating example to highlight the ideas behind using a multi-dimensional calibration score (Section 4.1):
>
> Assume a binary classification problem with $\mathcal{Y}=\{-1,1\}$ and prior probabilities $\mathbb{P}(Y=1)$ and $\mathbb{P}(Y=-1)$. The input $X$ is generated from a mixture of two Gaussians, with $\mathbb{P}(X|Y)=\mathcal{N}(Y,\sigma_y^2)$. In this example, we can compute the posterior $\mathbb{P}(Y|X)$ using Bayes rule. Consider the following three classifiers:
>
> (i) $\pi_0(x):=\mathbb{P}(y|x)$
>
> (ii)   $\pi_1(x) := \mathbb{P}(y|x) \text{ if } x<0$ and $(\epsilon,1-\epsilon) \text{ if } x>0$
>
> (iii)   $\pi_2(x) := (\epsilon,1-\epsilon) \text{ if } x<0$ and $\mathbb{P}(y|x) \text{ if } x>0$
>
> where $\epsilon\sim\mathcal{N}(0,1)$. Here, $\pi_0(x)$ represents the ideal classifier, while $\pi_1(x)$ and $\pi_2(x)$ are ideal classifiers over half the range of \(x\), but uninformative over the remaining half. Let $s_i(x,y)$ denote the nonconformity score computed over the $i$-th classifier, it is clear that performing conformal prediction over $s_0(x,y)$ would be the most efficient, however, using either $s_1(x,y)$ or $s_2(x,y)$ will lead to suboptimal results in the uniformative region. In this case, performing conformal prediction in the 2-dimensional space defined by $\mathbf{s}(x,y)=[s_1(x,y),s_2(x,y)]^T$ is advantageous as the individual scores provide complementary information for different ranges of the input $x$. This can be seen from Fig.2(b), where taking both axes into account can help to identify regions with high density of true points versus false points. Setting $\alpha=0.1$, we obtain the following average set sizes: $1.05$ for $s_0(x,y)$, $1.48$ for $s_1(x,y)$, $1.47$ for $s_2(x,y)$ and $1.24$ for our proposed method. Figure 2(c) compares the set sizes per $x$-domain. As expected, $s_0(x,y)$ obtains sets of size $1$ for the entire range, except for $x\in[-1.5,1.5]$, where the two Gaussians overlap. In contrast, $s_1(x,y)$ and $s_2(x,y)$ produce larger proportions of 2-element sets in the noisy regions, on the right for $s_1(x,y)$ and on the left for $s_2(x,y)$. Our method, utilizes both $s_1(x,y)$ and $s_2(x,y)$, obtaining similar behavior to the ideal case provided by $s_0(x,y)$.
>
> More details are provided in Appendix A.
>
> In general, the intuition behind our approach is that higher-dimensional spaces can better separate correct from incorrect labels, potentially leading to more efficient prediction sets with fewer false labels.

---

> ### Author Response · Authors · 2024-11-24
> **Response to Reviewer 52ya - Part II**
>
> > On the relation of the proposed method multi-score setting vs CP for ensemble models
>
> Thank you for pointing this out. The main idea in the paper is how we can use multiple nonconformity scores to improve CP efficiency, where we propose to perform the calibration in the high-dimensional space instead of averaging the scores and performing standard one-dimensional CP. This idea can be applied in the case we have multiple different scores, which can be obtained in different ways: standard ensemble of multiple different models, test-time augmentation where scores are computed for the output of different augmented versions of the input image, multiple types of scores computed from a single output (e.g. Thr, APS, RAPS, SAPS) and our proposed self-ensemble that extracts scores from multiple classification heads, trained with diversity-regularized loss. Our main results are shown for the self-ensemble approach, while we also provide results for the other two cases: test time augmentation and standard ensemble.
>
> Following your comment, we added results  for combining multiple types of scores. Specifically, we examined a setting where instead of considering multiple classification heads, we use a single head and compute different conformity scores: Thr, APS, RAPS and SAPS. Table D.2 summarizes the results for all datasets:
>
> | **α** | **Dataset** | Multi Score | Best Head | Uniform | Optimized | Norm |
> |-------|-------------|-------------|-----------|----------|-----------|------|
> | 0.1   | CIFAR100    | 33.90       | 35.12     | 35.32    | 34.96     | 35.54 |
> | 0.1   | TinyImageNet | 58.08      | 61.62     | 62.96    | 62.70     | 78.45 |
> | 0.15  | CIFAR100    | 18.33       | 27.88     | 28.12    | 27.52     | 28.18 |
> | 0.15  | TinyImageNet | 35.84      | 48.55     | 49.47    | 49.48     | 55.75 |
> | 0.2   | CIFAR100    | 8.25        | 9.34      | 9.19     | 8.35      | 13.02 |
> | 0.2   | TinyImageNet | 19.99      | 24.94     | 27.49    | 26.94     | 34.49 |
> | 0.01  | PathMNIST   | 6.26        | 6.69      | 6.67     | 6.5       | 6.56  |
> | 0.02  | PathMNIST   | 3.31        | 4.56      | 2.98     | 2.96      | 3.07  |
>
> We see that combining multiple scores improves the results compared to the best single score, and that the proposed method obtains the smallest prediction sets in almost all cases. The advantage of this score fusion is that it does not require any additional modifications to the model or further fine-tuning iterations.
>
> > On the extensiveness of the empirical comparison, and including ImageNet in the experiments
>
> Following your suggestion, we added an experiment on the ImageNet dataset to demonstrate the effectiveness of the proposed method with a large number of classes. The results are summarized in Table D.8 (Appendix D.4):
>
> | **α** | **Score** | Multi Score | Best Head | Uniform | Optimized | Norm |
> |-------|-----------|-------------|-----------|----------|-----------|------|
> | 0.1   | RAPS      | 4.1         | 4.36      | 13.47    | 14.63     | 13.47 |
> | 0.1   | SAPS      | 4.94        | 5.36      | 13.85    | 14.99     | 13.85 |
> | 0.2   | RAPS      | 1.82        | 2.4       | 2.1      | 2.1       | 2.1   |
> | 0.2   | SAPS      | 1.57        | 1.68      | 2.04     | 2.06      | 2.04  |
>
> Furthermore, we tested the multi-score approach on the 20 Newsgroups dataset to make the empirical comparison more extensive by considering different data domains, as suggested also by reviewer 9Lus. The results are summarized in Table
> D.7 (Appendix D.4):
>
> | **α** | **Score** | Multi Score | Best Head | Uniform | Optimized | Norm |
> |-------|-----------|-------------|-----------|----------|-----------|------|
> | 0.08  | RAPS      | 1.46        | 9.61      | 9.58     | 9.49      | 1.5  |
> | 0.08  | SAPS      | 1.47        | 10.66     | 2.09     | 2.18      | 2.35 |
> | 0.05  | RAPS      | 2.29        | 10.1      | 10.01    | 10.07     | 2.32 |
> | 0.05  | SAPS      | 1.97        | 10.9      | 2.4      | 2.46      | 2.62 |
> | 0.02  | RAPS      | 4.28        | 11.73     | 11.73    | 11.47     | 4.36 |
> | 0.02  | SAPS      | 3.28        | 11.57     | 3.45     | 3.34      | 3.65 |
>
> In both cases, we can see the benefit of the proposed method over the baselines.
>
> > On figures clarity and providing tables to show the results
>
> Thank you for your remark. Trying to replace Figures 3, D.1, D.2 and D.3. with tables resulted in overly large and confusing layouts, therefore the figures were enlarged instead for better clarity. We took your suggestion into account and replaced other experimental plots with tables throughout the paper.

---

> ### Author Response · Authors · 2024-11-24
> **Response to Reviewer 52ya - Part III**
>
> >  On putting some results from Appendix B in the main part
>
> Following the reviewers comments, we added new parts to the paper (a demonstrating example for motivation, and additional variants of our method) . In addition, the figures were enlarged for clarity. Thus, although we wanted to include some experiments from Appendix D in the main section, as you suggested, there was not enough space left.
>
> > On Figure 2 - are all the compared methods employed on the multi-head model trained by the loss in Equation (11)?
>
> Yes, all the compared methods on Figure 3 were employed on the multi-head model trained by the loss in Equation (11). We added results for a vanilla approach with a single head for comparison in Appendix D.5 (Table D.10).

---

> > ### Comment · Reviewer_52ya · 2024-12-01
> >
> > Thank you for the response. While the response addressed some of my concerns, the explanation of why the method work is not convincing without exact validation. Besides, the position of this work should be clarified in the paper, which may requires to update the manuscript a lot. Therefore, I will keep my score as reject this time.

---

### Official Review · Reviewer_3FMy · 2024-11-05

**Soundness:** 3
**Presentation:** 3
**Contribution:** 2
**Rating:** 6
**Confidence:** 4

**Summary:**

This paper proposes an extension to conformal prediction for classification tasks. Standard conformal prediction constructs prediction sets with guaranteed coverage by thresholding a nonconformity score, which measures the disagreement between the model and the ground truth class label. This work has two main contributions: (a) a variant of conformal prediction that is compatible with multidimensional nonconformity scores, and (b) a self-ensembling method for producing these multidimensional nonconformity scores using diverse classification heads. The proposed method partitions the multidimensional space and then selects a set of cells to include in prediction sets by thresholding the cells based on the relative frequency of false labels. The proposed approach is shown experimentally to lead to smaller prediction sets.

**Strengths:**

Strengths of the paper include the following:
- The main claims are supported with evidence. Theoretical justification for the proposed procedure is provided in Appendix A.1. Smaller prediction set sizes are also demonstrated in Figure 2 and Figure 3. However, see weaknesses below regarding multiple classification heads.
- There is a good amount of exploration performed, even beyond what is reported in the main paper. For example, results on standard ensembling, test-time augmentations, and size of $\mathcal{D}_\text{cells}$ are all useful.
- The paper is well-written. I thought the related work section in particular explored a good amount of relevant literature.

**Weaknesses:**

Weaknesses of the paper include the following:
- The empirical evidence provided in the paper is overall good, but the role of the multiple classification heads is still unclear. The best head baseline uses a network trained with multiple heads. How would the proposed approach compare to a vanilla conformal prediction baseline trained with a single head (i.e. the network prior to attaching the six additional classification heads)?
- The contribution of the paper is fairly solid in terms of theoretical and empirical justification. However, from a practical perspective, it seems more likely that rather than investing time into using multidimensional nonconformity scores, a practitioner would likely simply focus on improving the underlying model generating the nonconformity scores. The cases in which the multidimensional approach succeed are those where the model is suboptimal, and fixing the model will remove the need to use multidimensional scores. This limits the significance of the proposed work.
- The benefits of the proposed approach are empirical in nature. There does not seem to be a way to know beforehand whether multidimensional nonconformity scores will help or not. In some sense, this is a modeling question that depends on the distribution of nonconformity scores in the multidimensional space. There will be some datasets where the modeling assumptions proposed here (analogous to a KNN variant with $K=1$) will be beneficial, and others where it will not be.

**Questions:**

- Please see the first point under the weaknesses section about. How would the proposed approach compare to a vanilla RAPS/SAPS baseline with one classification head (i.e. before the six additional heads are attached)?
- In Figure 4, the nonconformity scores appear to be distributed in a pattern resembling a square. Would an L1 norm perform better than an L2 norm in this case?
- What is the effect on the algorithm as the dimensionality increases? Is there a curse of dimensionality effect here?
- In Algorithm 1 (line 6), when do the duplicate cells occur?

Minor comments:
- The cells in Figure 1 are difficult to understand.
- Figures 2 and 3 are small and therefore hard to read.
- Since Section 3.1 discusses background work, it should probably be moved out of Section 3, which is title "Proposed Method".

---

> ### Author Response · Authors · 2024-11-24
> **Response to Reviewer 3FMy - Part I**
>
> Thank you for your helpful and constructive review.
>
> > On the comparison to a vanilla conformal prediction baseline trained with a single head.
>
> Following your comment, we added an experiment using a vanilla conformal prediction baseline in Appendix D.5. In the vanilla baseline, the conformal prediction procedure is performed over the original classification head, without the addition of multiple classification heads. We observe that the results are similar to the Best Head baseline (Table D.10):
>
> | **α**  | Multi Score | Vanilla RAPS | Best Head |
> |--------|-------------|--------------|-----------|
> | **0.1**| 15.32       | 39.27        | 35.29     |
> | **0.2**| 2.33        | 8.09         | 9.37      |
>
>
> >On the practical usage - instead of using multidimensional nonconformity scores, a practitioner would likely focus on improving the underlying model generating the nonconformity scores.
>
> Thank you. It is true that improving the underlying model would, in general, contribute to improving the efficiency of the conformal prediction procedure, and this will be a main focus of the practitioner to first try improving the underlying model. However, our proposed method presents an additional avenue for improvement that is orthogonal to the improvement of the underlying model. In many practical cases, the search for a better architecture that yields preferable performance is limited by the size or the quality of the data, as well as cost considerations that often impose constraints on model size and inference time. Our approach proposes to enhance the efficiency of any given model with a small addition of few classification heads, or by utilizing an already available ensemble of several models. This approach of improving the efficiency of conformal prediction over a given model was investigated in other works, as detailed in the related work section “Combining nonconformity scores”.
>
> To further strengthen this claim we conducted an additional study where we used an improved model  - ViT instead of ResNet50. We show that even for the stronger model we can improve conformal prediction efficiency, as can be seen from Table D.6 :
>
> | **Model**        | **Acc.** | **Score** | Multi Score | Best Head | Uniform | Optimized | Norm  |
> |-------------------|----------|-----------|-------------|-----------|---------|-----------|-------|
> | **ViT**          | 0.8      | **RAPS**  | 4.34        | 22.11     | 16.21   | 15.93     | 5.51  |
> |                  |          | **SAPS**  | 2.33        | 12.77     | 2.43    | 2.45      | 2.60  |
> | **ResNet50**     | 0.69     | **RAPS**  | 15.32       | 35.29     | 24.37   | 22.47     | 22.49 |
> |                  |          | **SAPS**  | 15.46       | 37.41     | 22.05   | 21.80     | 22.79 |
>
> > On the benefit of the proposed method for different datasets, and the dependency on the distribution of nonconformity scores in the multidimensional space.
>
> Thank you for pointing this out. Indeed, the benefit of the proposed method relies on the distribution of the nonconformity scores in the multidimensional space. We added a synthetic example (Section 4.1) that demonstrates that in case the nonconformity scores are noisy for part of the input space, we can benefit from considering multiple scores instead of a single one. This aligns with previous findings, demonstrating that a weighted combination of various scores enhances the efficiency of conformal prediction. In this paper, we extend these ideas and treat the case of multiple scores in a more general way, inspecting it as a multidimensional score and performing the selection in the high dimensional space, while taking into account the distribution of true versus false labels. Our extensive evaluations over different datasets, scores, coverage levels, and data sizes, give strong evidence for the benefit of our method under diverse settings. Moreover, we show that we can modify the distribution of scores using temperature scaling in order to obtain better separation between true and false scores (see Figs. D.5 and D.6).

---

> ### Author Response · Authors · 2024-11-24
> **Response to Reviewer 3FMy - Part II**
>
> > On the difference between L1 and L2 norm baselines
> Following tour comment, we added an experiment comparing L1 and L2 baselines in Appendix D.5. Table D.9 summarizes the results:
>
> | α | Score | L1 | L2 | Multi Score |
> |---|-------|-----|----|----|
> | 0.1 | RAPS | 22.98 | 22.49 | 15.32 |
> | 0.1 | SAPS | 22.25 | 22.79 | 15.46 |
> | 0.2 | RAPS | 6.75 | 7.51 | 2.33 |
> | 0.2 | SAPS | 7.62 | 7.85 | 2.36 |
>
> We observe that both baselines obtain similar performance.  We can explain this by noting that the resemblance to the square-like appearance of the nonconformity scores is more prominent at lower scores. The desired coverages we aim for requires a higher threshold, and as a result, the difference between the L1 and L2 norms becomes less prominent. This is further illustrated in Fig. 4, where it is evident that the L2 norm threshold is far from the square-like distribution.
>
> > On the effect of increasing dimensionality
>
> Our experimental results include different datasets with varied numbers of classes (between 9 and 1000) and different sizes of calibration points. We do not observe any strong effect in terms of the dimensionality of the problem or the data. In terms of the score dimensionality, we can see that the efficiency improves with increasing the number of classification heads, appearing to plateau around four to five heads, with occasional slight increases thereafter. This behavior might be linked to the curse of dimensionality, suggesting that the current approach is well-suited for moderately sized score spaces. From a practical standpoint, this aligns with typical scenarios, as ensembling methods are generally constrained to combining only a few models due to size and cost considerations.
>
> > On the case of duplicate cells (Algorithm 1, line 6)
>
> It might happen that two or more points have the same score, thus can cause duplicate cells. In practice, this rarely occurs in dimensions higher than 1. In such cases, we remove the duplicate cells but account for the multiplicity in the cell score computation in Eq. (9).
>
> > On the cells in figure 1
>
> Following your comment, we updated the figure to improve clarity, zooming in to the right bottom corner.
>
> > On increasing the size of Figures 2 and 3
>
> Thank you for your remark. As you suggested, the figures have been enlarged.
>
> > On moving the part on background work to another section
>
> We moved the part of the conformal prediction background to a dedicated section.

---

> ### Author Response · Authors · 2024-12-01
>
> Dear reviewer 3FMy,
>
> Thank you very much again for your helpful and constructive feedback. Your insights have been invaluable in refining our work, and we are grateful for the time and effort you have dedicated to reviewing our manuscript.
>
> As the review process deadline approaches, we kindly request that you revisit our submission in light of the revisions and corrections we have made in response to your valuable feedback. We have carefully addressed the points you raised and believe these changes have strengthened the paper significantly.
>
> If you find the revisions satisfactory, we would greatly appreciate it if you could consider raising your score to reflect the improvements.
> We would be more than happy to provide further clarifications if you have any additional concerns.
>
> Best,
> Authors

---

> > ### Comment · Reviewer_3FMy · 2024-12-02
> >
> > Thanks to the authors for their response. The new results showing performance of a vanilla network, L1, and alternate architectures are valuable, and show that the proposed approach can still produce smaller prediction set sizes even in these cases. I have increased my score accordingly.

---

### Official Review · Reviewer_4GpN · 2024-11-12

**Soundness:** 4
**Presentation:** 4
**Contribution:** 3
**Rating:** 6
**Confidence:** 4

**Summary:**

This paper studies a multi-dimensional non-conformity approach to conformal prediction. They introduce a novel method to identify high probability regions with low low false-to-true labels ratio, which requires to carefully design a calibration procedure with multiple stages, including partitioning the space, ranking different partitions, and selecting the high quality partitions.

**Strengths:**

This paper is very well-written and has a very clear flow. The multi-dimensional CP looks like a promising avenue of research. They provide a novel perspective of finding conformalized regions by ranking the false-to-true ratio and they also show sufficient experimental results to back up their claim about improving efficiency.

**Weaknesses:**

I believe this is a very good paper. Although I do have some suggestions to further improve both the theoretical and algorithmic perspectives.

- You break the dataset in two parts and then use the second part as a recalibration to ensure coverage. A natural way to improve the efficiency of your method and use all the data together is to use a leave-one-out method. At each time, you can use all the data but one (say X_i, Y_i), go through the partitioning and ranking and then find the smallest threshold (rank) such that the selected regions include the left-out sample (X_i, Y_i). Lets call that threshold (rank) tau_i. Then you can take (1-\alpha))-quantile of the ranks and use that as the final threshold (rank) to make the regions. There are some subtleties to make this work, you might want to look at https://arxiv.org/abs/1905.02928 for more details.

- For the case of scalar scores there are some theoretical underpinnings that shows the optimal (think of it as the tightest possible) prediction sets are of the form of level-sets of a scalar function. (i.e., thresholding a scalar score) These results are considered classic in CP (look at https://www.stat.cmu.edu/~jinglei/LeiW14.pdf) and they have been also recently generalized to the more advanced case of CP with conditional coverage requirements (look at https://arxiv.org/pdf/2406.18814). These results and their insights also heavily contributed in the line of research around designing better scores. Therefore, I believe it is important to discuss whether a multi-dimensional score design has any fundamental theoretical underpinnings or not. For instance, it would be nice to think of a scenario where (in the population regime) the optimal prediction sets are actually characterized by a multi-dimensional score. Or maybe trying to show that in a (perhaps simplified) scenario the multi-dimensional perspective is provably a better approximation of the optimal prediction sets. These kind of results and discussions are more than just an effort for having some theory in the paper, but rather can bring useful insights for follow-up works in this line of research.

I am willing to increase my score in case of a satisfactory response.

**Questions:**

I have no further questions. Might ask more once I see the authors response.

---

> ### Author Response · Authors · 2024-11-24
> **Response to Reviewer 4GpN**
>
> Thank you for your helpful and constructive review.
>
> >  On a leave-one-out version of the proposed method
>
> Thank you very much for your helpful suggestion. We implemented a variant of your suggested approach following [1] and [2]. We added the exact implementation in Algorithm B.1. The core idea is to consider the entire calibration data but each time exclude the $i$th point from the set of centers and from the score computation in Eq. (9). We sweep over all possible labels $y\in\mathcal{Y}$ and assign it to the closest center while removing the $i$th center. We perform a similar assignment for the $i$th score $s(X_i,Y_i)$. Then, we compare the rank of the chosen cells, and include $y$ in the final prediction if its rank is smaller than $(1-\alpha)(m + 1)$ hold-out ranks evaluated on the true labeled data for every $i\in\\{1,\ldots,m\\}$. We examined the performance of the Jacknife+ and the split approaches. In both settings, the required 1-$\alpha$ coverage is achieved, but the Jackknife+ version obtained smaller sets, as expected:
>
> | α | Score | Multi Score Jackknife+ | Multi Score |
> |---|-------|----------------------|-------------|
> | 0.1 | RAPS | 14.85 | 15.32 |
> | 0.1 | SAPS | 14.61 | 15.46 |
> | 0.2 | RAPS | 2.26 | 2.33 |
> | 0.2 | SAPS | 2.22 | 2.36 |
>
> However, the improvement appears to be small in this case and may not justify the additional computational cost.
>
> [1] Barber, R. F., Candes, E. J., Ramdas, A., & Tibshirani, R. J. (2021). Predictive inference with the jackknife+.‏
>
> [2] Romano, Y., Sesia, M., & Candes, E. (2020). Classification with valid and adaptive coverage. Advances in Neural Information Processing Systems, 33, 3581-3591.
>
> >  On the theoretical underpinnings of the multi-dimensional score.
>
> Thank you very much for your important insights. We can indeed think of scenarios where the optimal prediction sets are actually characterized by a multi-dimensional score. We added such an example to the paper, along with figures for illustration (Figures 2 and A.1.):
>
> Assume a binary classification problem with $\mathcal{Y}=\\{-1,1\\}$ and prior probabilities $\mathbb{P}(Y=1)$ and $\mathbb{P}(Y=-1)$. The input $X$ is generated from a mixture of two Gaussians, with $\mathbb{P}(X|Y)=\mathcal{N}(Y,\sigma_y^2)$. In this example, we can compute the posterior $\mathbb{P}(Y|X)$ using Bayes rule. Consider the following three classifiers:
>
> (i) $\pi_0(x):=\mathbb{P}(y|x)$
>
> (ii)   $\pi_1(x) := \mathbb{P}(y|x) \text{ if } x<0$ and $(\epsilon,1-\epsilon) \text{ if } x>0$
>
> (iii)   $\pi_2(x) := (\epsilon,1-\epsilon) \text{ if } x<0$ and $\mathbb{P}(y|x) \text{ if } x>0$
>
> where $\epsilon\sim\mathcal{N}(0,1)$. Here, $\pi_0(x)$ represents the ideal classifier, while $\pi_1(x)$ and $\pi_2(x)$ are ideal classifiers over half the range of \(x\), but uninformative over the remaining half. Let $s_i(x,y)$ denote the nonconformity score computed over the $i$-th classifier, it is clear that performing conformal prediction over $s_0(x,y)$ would be the most efficient, however, using either $s_1(x,y)$ or $s_2(x,y)$ will lead to suboptimal results in the uniformative region. In this case, performing CP in the 2-dimensional space defined by $\mathbf{s}(x,y)=[s_1(x,y),s_2(x,y)]^T$ is advantageous as the individual scores provide complementary information for different ranges of the input $x$. This can be seen from Fig.2(b), where taking both axes into account can help to identify regions with high density of true points versus false points. Setting $\alpha=0.1$, we obtain the following average set sizes: $1.05$ for $s_0(x,y)$, $1.48$ for $s_1(x,y)$, $1.47$ for $s_2(x,y)$ and $1.24$ for our proposed method. Figure 2(c) compares the set sizes per $x$-domain. As expected, $s_0(x,y)$ obtains sets of size $1$ for the entire range, except for $x\in[-1.5,1.5]$, where the two Gaussians overlap. In contrast, $s_1(x,y)$ and $s_2(x,y)$ produce larger proportions of 2-element sets in the noisy regions, on the right for $s_1(x,y)$ and on the left for $s_2(x,y)$. Our method, utilizes both $s_1(x,y)$ and $s_2(x,y)$, obtaining similar behavior to the ideal case, provided by $s_0(x,y)$.
>
> More details are provided in Appendix A.

---

> > ### Comment · Reviewer_4GpN · 2024-11-25
> >
> > I have read the authors response. It is addressing most of my comments. However, I still believe a direct discussion of length improvement of multi dimensional scores from the perspective of precise characterization of optimal prediction set size (look at  https://www.stat.cmu.edu/~jinglei/LeiW14.pdf and https://arxiv.org/pdf/2406.18814) is missing in the paper. These works provide a very foundational characterization of length optimality in terms of level sets of some density functions (a scalar score with a scalar threshold). Now, the question is how this multi dimensional perspective is improving the set size. To me, it should be more of a finite sample effect, rather than a fundamental gain in population regime. Anyhow, this should be discussed in the paper. I find the example helpful, but a direct discussion with the existing results on length optimality can cover this hole. Addressing that, I believe the contribution of the paper is enough for acceptance.

---

> ### Author Response · Authors · 2024-11-27
> **Thanks for your response - Part I**
>
> Thank you very much for reviewing our response and for your feedback.
>
> Following your suggestion, we have included in Appendix A.1. a more detailed discussion on how the multi-dimensional perspective enhances set size optimization, relating to the above mentioned references:
>
> "When $\mathbb{P}(Y|X)$ is known, it was shown that the optimal set with minimal size under coverage constraint is given by $\Gamma^*(x)=\\{y\in\mathcal{Y}|p(y|x)>q_\alpha\\}$  (Lei & Wasserman, 2014; Sadinle et al., 2019; Kim et al., 2021). This implies that the optimal set is a level set of the distribution $p(y|x)$. Thus, in our example, thresholding $s_0(x,y)=1-p(y|x)$ results in the optimal set. Using only $s_1(x,y)$ and $s_2(x,y)$ the optimal set can be equivalently defined as:
> \begin{align}
> \Gamma^*(x)&=\\{y\in\mathcal{Y}|(\mathbf{1}\\{x\leq 0\\}\cdot s_1(x,y)+\mathbf{1}\\{x>0\\}\cdot s_2(x,y))<1-q_\alpha\\} =\\{y\in\mathcal{Y}|\mathbf{s}^T(x,y)\cdot\mathbf{i}_x(x)<1-q_\alpha\\},
> \end{align}
> where $\mathbf{i}_x(x)=[\mathbf{1}\\{x\leq 0\\}, \mathbf{1}\\{x>0\\}]^T$. Thus, we obtain that the optimal set is a function of the 2-dimensional nonconformity score $\mathbf{s}(x,y)$. In this example, each classifier specializes on a different subdomain of the input space $\mathcal{X}$. Another practical case is when classifiers specialize on different parts of the output space $\mathcal{Y}$. For example, consider $\mathcal{Y} = \\{0, 1, 2\\}$, and the following three classifiers:
>
> $\pi_a(x) := \mathbb{P}(y=a|x)  \text{ if } y=a$, or $\epsilon \text{ if } y=(a+1) \text{ mod } 3$ or $1-\mathbb{P}(y=a|x)-\epsilon, \text{ if } y=(a+2) \text{ mod } 3,\text{ } a\in\\{0,1,2\\},$
>
> where $\epsilon\sim\mathcal{N}(0,1)$. In this case, the optimal set is given by:
>
> $\Gamma^*(x)=\\{y\in\mathcal{Y}|\mathbf{s}^T(x,y)\cdot\mathbf{i}_y(y)<1-q\\}$
>
> where $\mathbf{i}_y(y)=[\mathbf{1}\\{y=1\\}, \mathbf{1}\\{y=2\\},\mathbf{1}\\{y=3\\}]^T$.
>
> We conclude that whenever $p(y|x)=\phi(\mathbf{s}(x,y);x,y)$, where $\phi:\mathcal{S}\times\mathcal{X}\times\mathcal{Y}\rightarrow [0,1]$ is a non-degenerate function of the multi-score vector, the optimal set relies on $\mathbf{s}(x,y)$, i.e.:
>
> $\Gamma^*(x)=\\{y\in\mathcal{Y}|\phi(\mathbf{s}(x,y);x,y)<1-q_\alpha\\}\text{                         }$            (17)
>
> In contrast, relying solely on a single score will result in a suboptimal solution. Note that, according to Eq. (17), the ideal set corresponds to a level set of $\phi(\mathbf{s}(x, y); x, y)$, rather than $\mathbf{s}(x, y)$ itself. This implies that, in general, the decision boundaries in the multi-dimensional score space can be arbitrarily complex, depending on the properties of $\phi$.
>
> In practice, we have access to neither the conditional distribution $\mathbb{P}(Y|X)$ nor the mapping function $\phi$. Instead, we aim to solve the problem of minimizing the set size subject to a coverage constraint, similarly to (Stutz et al., 2021; Bai et al., 2021; Kiyani et al., 2024). Since the space of all possible prediction sets is overly complex, the problem must be relaxed. Bai et al., 2021 proposed to optimize an arbitrary class
> of prediction sets $\Gamma_{\theta}$ parametrized by $\theta$, while Stutz et al., 2021 optimize a parametrized score $s_\theta(x,y)$. In (Kiyani et al., 2024), structured prediction sets of the form $\Gamma_h^s(x) = \\{y \in Y | s(x, y) \leq h(x)\\}$ were considered, where $h:\mathcal{X}\rightarrow \mathbb{R}$ is a learned adaptive threshold. In contrast, we work in the multi-score domain and consider sets defined as a union of a subset $I\subseteq 2^k$ of cells in $D_\textrm{cells}$, i.e. $\Gamma_{I}^\mathbf{s}(x)=\\{y\in\mathcal{Y}|\mathbf{s}(x,y)\in\cup_{i\in I}\mathcal{C}_{i}\\}$. Accordingly, the relaxed optimization problem can be written as:
>
> $arg\min_{I\subseteq 2^k} \mathbb{E}\left[\textrm{size}(\Gamma_{I}^\mathbf{s}(X))\right] $
>
> $\text{s.t. } \mathbb{E}\left[\mathbf{1}\\{Y\in \Gamma_{I}^\mathbf{s}(X)\\}\right]\geq 1-\alpha$
>
>
> where $\textrm{size}(\Gamma_{I}^\mathbf{s}(X))=\sum_{q=1}^Q\mathbf{1}\\{Y\in\Gamma_{I}^\mathbf{s}(X)\\}=\sum_{i\in I}\sum_{q=1}^Q\mathbf{1}\\{Y\in\mathcal{C}_{i}(X)\\}$.
>
> A finite sample approximation of the expected set size is given by:
>
> $\mathcal{E}=\frac{1}{k}\sum_{j=1}^k\sum_{i\in I}\sum_{q=1}^Q \mathbf{1}\\{\mathbf{s}(X_j, q) \in \mathcal{C}_i\\}.$
>
> Note that this is equivalent to summing the cell scores $D_i$  defined in Eq. (9) (without removing duplicate cells):
>
> $\mathcal{E} = \frac{1}{k}\sum_{i\in I}D_i.$
>
> Therefore, solving the optimization problem in Eq. (18) does not require enumerating all possible sets $ I $, which is computationally infeasible. Instead, we can rank the cells according to $D_i$ and take the $(1-\alpha)$ proportion of cells with the lowest score values. To obtain exact coverage we perform a recalibration over $\mathcal{D}_\textrm{re-cal}$.

---

> > ### Author Response · Authors · 2024-11-27
> > **Thanks for your response - Part II**
> >
> > We conclude that in practical scenarios, where a single score does not provide the full information on the conditional distribution $\mathbb{P}(y|x)$, we benefit from using a multi-dimensional score $\mathbf{s}(x,y)$. It may appear that optimizing for set size efficiency in the multi-score space exponentially increases the number of possible prediction sets to be considered, which makes the optimization more challenging compared to the single-dimensional case. However, our cell partitioning and ranking procedure relaxes the problem to a convenient structured prediction with a simple selection rule that does not require any iterative optimization procedures. Note that the number of cell centers and the summation operation over all scores that fall in the chosen region, remain fixed regardless of the dimensionality of $\mathbf{s}(x,y)$. However, as $n$ increases the cells move apart from each another, when the scores are nonidentical and provide complementary information. Moreover, if each dimension contributes information about the actual conditional distribution $\mathbb{P}(Y|X)$, we anticipate an improved separation between true and false scores. Consequently, the selected subset of cells is expected to exhibit lower $D_i$ values, leading to smaller prediction sets. This is demonstrated in Fig. A.2. presenting the distribution of $D_i$ for the chosen cells. We observe that as $n$ increases, the values of $D_i$ become smaller."
> >
> > -----------------------------------------------------------------------------
> >
> > Jing Lei and Larry Wasserman. Distribution-free prediction bands for non-parametric regression.
> > *Journal of the Royal Statistical Society Series B: Statistical Methodology*, 76(1):71–96, 2014.
> >
> > Mauricio Sadinle, Jing Lei, and Larry Wasserman. Least ambiguous set-valued classifiers with bounded error levels.
> > *Journal of the American Statistical Association*, 114(525):223–234, 2019.
> >
> > Shayan Kiyani, George J Pappas, and Hamed Hassani. Length optimization in conformal prediction.
> > *In The Annual Conference on Neural Information Processing Systems*, 2024.
> >
> > David Stutz, Krishnamurthy Dj Dvijotham, Ali Taylan Cemgil, and Arnaud Doucet. Learning optimal conformal classifiers. *In International Conference on Learning Representations*, 2021.
> >
> > Yu Bai, Song Mei, Huan Wang, Yingbo Zhou, and Caiming Xiong. Efficient and differentiable conformal prediction with general function classes. *In International Conference on Learning Representations*, 2021.

---

> ### Comment · Reviewer_4GpN · 2024-11-29
>
> I have read the authors response and other reviewers opinions. I have no further concerns and vote for acceptance.

---

### Author Response · Authors · 2024-11-24
**General Comment**

We would like to thank the reviewers for their time and effort in reviewing our paper. Their constructive comments helped us to improve the paper as reflected in the revised manuscript. Following their comments, we made major changes to the paper's content, clarifying important aspects, and providing additional details on the experimental setup. Moreover, we added new experiments and ablation studies that strengthen the empirical verification of our method.

We would like to emphasize the major modifications to the revised manuscript:

(i) We introduced an illustrative example that demonstrates the motivation for our proposed multi-dimensional approach (in Section 4.1).

(ii) We added two additional versions of our proposed method (Section 4.2): (1) A statistically efficient version based on a leave-one-out procedure, known as Jackknife+. (2) A soft version that considers for each point $b$ neighboring cells, where our core method is a private case with $b=1$.

(iii) We conducted extensive additional experimental studies, showing the robustness of the  proposed method with respect to: the data domain and size, the size of the calibration data and the number of center points, the strength of the diversity regularization and the choice of the nonconformity scores. These results further emphasize the advantage of the proposed method under varied settings.

---

### Author Response · Authors · 2024-11-28
**General response to all reviewers**

We sincerely appreciate the thoughtful and constructive feedback provided by all the reviewers. In response, we have uploaded a significantly revised version of our paper, which carefully addresses the review comments. We believe these revisions have led to meaningful improvements in several key aspects, thanks to your valuable suggestions.

Please do not hesitate to share any additional questions or concerns.

Thank you once again for your time, understanding, and insightful feedback.

---

### Meta-Review · Area_Chair_NqEE · 2024-12-21

**Metareview:**

This paper extends the standard approach for conformal prediction to produce prediction sets (uses a single non-conformity score) to multi-dimensional non-conformity scores. The key idea behind the approach is to partition the multi-dimensional space and selecting the high-quality partitions with low false-to-true ratio of labels using the calibrated thresholds. The paper provides theoretical analysis and empirical evidence for smaller prediction sets from the proposed method.

The reviewers' were generally positive about the paper, but also raised a number of questions. The author rebuttal was thorough and answered almost all questions satisfactorily, and the paper is also revised. One reviewer argued for rejection for the following reason: "the explanation of why the method works is not convincing without exact validation". This seems unfair to me given the motivation and set-size optimization perspective, and empirical results. As reviewer UGXT suggested, showing that the rankings are more aligned with ground truth aleatoric uncertainty using synthetic experiments can be helpful and I encourage the authors' to consider it.

Therefore, I recommend accepting the paper and strongly encourage the authors' to incorporate all the discussion in the camera copy to further improve the paper.

**Additional Comments On Reviewer Discussion:**

One reviewer argued for rejection for the following reason: "the explanation of why the method works is not convincing without exact validation". This seems unfair to me given the motivation and set-size optimization perspective, and empirical results. Another reviewer also supports my view.

---

### Decision · Program_Chairs · 2025-01-22

Accept (Poster)